



# Changes in the tropical upper tropospheric zonal momentum balance due to global warming

Abu Bakar Siddiqui Thakur[1,2] and Jai Sukhatme[1,2]

[1]Centre for Atmospheric and Oceanic Sciences, Indian Institute of Science, Bangalore 560012, India
[2]Divecha Centre for Climate Change, Indian Institute of Science, Bangalore 560012, India

**Correspondence:** Abu Bakar Siddiqui Thakur (thakur.abubakar@gmail.com)

**Abstract.**

The zonal momentum budget of the deep upper tropics is studied in the context of present and future climates. In the zonal mean, as is known, a robust balance exists between the acceleration by the horizontal eddy momentum flux convergence and the deceleration by the mean meridional momentum advection. During summer, climatological stationary Rossby waves over the Asian monsoon longitudes converge westerly momentum into the tropics and are the primary contributors to the eddy term. During winter, anomalous westerly winds over the tropical East Pacific allow extratropical waves to propagate into the deep tropics, where they tend to break and decelerate the flow. When integrated over all longitudes, eddies from these two regions sum constructively in summer and destructively in winter, always yielding a net positive momentum forcing that balances the mean flow term. The state-of-the-art CMIP6 suite qualitatively captures these features in the historical run and shows that the momentum fluxes change due to global warming. In summer, stationary eddy circulations in the Asian monsoon zone weaken in the upper troposphere (UT) but strengthen in the lower stratosphere (LS). Greater upward mass flux from the troposphere forces a stronger divergence and a more intense circulation in the LS following a Sverdrup vorticity balance. This strengthening of summertime tropical and subtropical stationary waves in the LS is observed over all longitudes and is verified in an idealized aquaplanet general circulation model experiment. In winter, eddy westerlies over the East Pacific longitudes decrease in strength due to the expected weakening of the subtropical stationary waves with warming. This causes a significant decrease in the propagation of extratropical waves into this region, along with a drop in eddy potential vorticity fluxes associated with these waves. Thus, apart from the mean meridional flux, which weakens due to the projected weakening of the Hadley Cells, our analysis of warming simulations clearly suggests significant and robust changes in the eddy momentum fluxes in the deep tropics. Potential implications of these changes in the context of the zonal mean flow and regional circulations are discussed.

*Keywords: tropical eddy and mean flow, zonal momentum, tropical dynamics, present climate, climate change, CMIP6*





# 1 Introduction

The zonally averaged zonal momentum budget of the tropical upper troposphere has been the subject of numerous investigations (for example, Lee, 1999; Dima et al., 2005; Kraucunas and Hartmann, 2005). Reanalysis data shows that, in the present-day climate, the primary balance is between the acceleration of the zonal mean flow by eddies and the deceleration provided by the seasonally reversing Hadley circulation. While climatological stationary Rossby waves and tropical transients like the MJO converge momentum into the tropical latitudes, the Hadley circulation advects momentum out of the tropics, offsetting the eddy acceleration and decelerating the flow (Dima et al., 2005; Lee, 1999). In the annual mean, the deceleration by the mean meridional circulation exceeds the eddy momentum flux convergence from tropical and extratropical sources and leads to tropical upper-tropospheric easterlies (Lee, 1999). There is a strong sense of anti-correlation between these two terms, even on seasonal timescales (Dima et al., 2005; Kelly and Mapes, 2011). In fact, Shaw (2014) noted a similar balance while emphasizing the roles of planetary and sub-planetary scale eddies in the winter-summer transition of the Northern Hemisphere (NH) circulation.

While a zonal mean perspective is useful, however, longitudinal asymmetries are inherent characteristics of the Earth system, particularly in the tropics with oceans and landmasses, monsoon regions, and deserts in the same latitude band. These zonal asymmetries are associated with interesting dynamics that are masked by the zonal averaging. For example, the monsoon-desert mechanism (Rodwell and Hoskins, 1996), wherein diabatic heating associated with intense tropical convection forces moist ascent over the Asian monsoon region and corresponding adiabatic descent of dry air over the eastern Sahara and the Mediterranean leading to desertification of these regions. Further, circulations that average to give the zonal mean Hadley circulation are confined to specific regions where both synoptic and intraseasonal activity are important (Hoskins et al., 2020; Hoskins and Yang, 2021, 2023), and these can be isolated from the divergent component of the horizontal flow (Keyser et al., 1989). In fact, contrary to the zonally symmetric view of the Hadley circulation, Galanti et al. (2022) show that the meridional overturning circulation organizes in clusters with variability on interannual timescales (see also, Sun et al., 2019) and Raiter et al. (2020) present a large-scale conveyor belt view of the tropical circulation. Further, the long-term trends of these longitudinally localized meridional overturning circulations exhibit large regional differences (Schwendike et al., 2014).

In the context of global warming, atmospheric circulation changes are a major source of uncertainty in modern-day climate projections (Shepherd, 2014). On regional scales, effects on the hydrological cycle are closely associated with changes in the tropical circulation (Collins et al., 2013; Vallis et al., 2015; Ma et al., 2018). Using thermodynamic arguments, Held and Soden (2006) showed that the increase in globally averaged surface temperatures must be accompanied by a weakening of the circulation (see also Schneider et al., 2010). The weaker circulation is also accompanied by a widening of the tropics, evidence of which has been found in observations (Hu and Fu, 2007; Seidel et al., 2008; Nguyen et al., 2013), full-complexity and idealized models (Lu et al., 2007; Frierson et al., 2007; Levine and Schneider, 2015). The Hadley cell widens with decreasing meridional temperature gradient (Adam et al., 2014), changes in the subtropical static stability (Frierson et al., 2007), and changing midlatitude baroclinic eddy activity (Chen et al., 2008). Such changes in the latitudinal extent of the tropics imply modifications in local radiative balances, changing precipitation patterns that induce shifts in regional climates and imply



drastic impacts to regional ecosystems. However, most studies have analyzed these changes only in the zonal mean. But, warming-related expansion may not be similar across longitudes (Nguyen et al., 2018; Staten et al., 2019). This is embodied in the pronounced longitudinal patterns in SST and precipitation that emerge in response to a spatially uniform increase in $CO_2$ (Xie et al., 2010). Hence, developing a deeper understanding of the zonal momentum balance becomes important since this is intimately tied to the local atmospheric circulations, and there is a possibility that the existing balance may change due to warming.

Further, many climate models develop an El Niño-like sea surface temperature (SST) anomaly in the equatorial Pacific in response to global warming (DiNezio et al., 2009; Xie et al., 2010). Paleoclimate evidence suggests that a similar El Niño-like SST pattern may have existed in past warm climates (Wara et al., 2005; Fedorov et al., 2006). Similar to the zonal mean effects of warming, the El Niño Southern Oscillation drives hemispherically symmetric climate variations (Seager et al., 2003), although the atmospheric response is different in the two cases (Lu et al., 2008; Tandon et al., 2013). The future state of the tropical Pacific SST has a strong bearing on the climate system because it may reorganize convection in the deep tropics and reshape the Hadley and Walker circulations. A stronger east-west temperature contrast in the tropical Pacific may trigger planetary-scale Rossby waves that cause a localized westerly acceleration and, in an extreme case, cause equatorial superrotation (Pierrehumbert, 2000; Tziperman and Farrell, 2009). Such waves may also contribute towards high-latitude warming through poleward heat fluxes (Lee et al., 2011). In fact, global climate model simulations suggest that the upper tropospheric winds may have transitioned to weak equatorial superrotation in past warm periods (Lan et al., 2023). Further, increased intraseasonal activity of the Madden-Julian Oscillation and the associated eddy fluxes may push the zonal mean flow towards a state of equatorial superrotation (Lee, 1999; Caballero and Huber, 2010; Arnold et al., 2012; Carlson and Caballero, 2016). Again, in the case of extreme equatorial superrotation where the upper tropospheric westerlies extend down to the surface, the equatorial Pacific SST gradient can weaken, leading to an El Niño-like state (Tziperman and Farrell, 2009). Such ideas have been explored previously using idealized models (Caballero and Carlson, 2018). However, to the best of our knowledge, these ideas are yet to be explored using state-of-the-art model simulations from the IPCC archive.

In this study, we explore the longitudinally dependent momentum budget in the upper tropical troposphere in present-day and future climates by using reanalysis and the CMIP6 suite of warming simulations. There are five sections in the paper. Section 2 contains a description of the data and techniques used in the study. Section 3 highlights that the present-day balance in the tropics is a little more delicate than what has been emphasized in the literature. We use reanalysis data and showcase the longitudinal structure of momentum fluxes as a function of time through the year. We then examine if the historical runs in CMIP6 agrees with present-day reanalysis; indeed, a qualitative agreement allows us to proceed and examine changes in fluxes with warming. In Section 4, we study these changes in the various flux terms with CMIP6 projections under global warming. We discuss statistically significant changes in the equatorial mean meridional and eddy fluxes during winter and summer by comparing the historical runs to a warming scenario. Changes in each season, their geographical origins, and possible causes are then brought forth. Section 5 contains a summary of the results and their discussion.



## 2 Data & Methods

### 2.1 Data

#### 2.1.1 Present-day climate

The data used in the first part of this study comprises daily-averaged horizontal winds at a resolution of $2.5°$ across 17 pressure levels from ERA5 (Hersbach et al., 2020) over 43 years from 1979 to 2021. We also make use of monthly mean GPCP Precipitation data (Adler et al., 2003), provided by NOAA (https://psl.noaa.gov/), with the same resolution and over the same period.

#### 2.1.2 Future climate

For estimates of the future climate, we make use of full complexity model simulations from the CMIP6 archive (Eyring et al., 2016, https://esgf-node.llnl.gov/projects/cmip6/). We employ 23 model simulations forced using the Shared Socioeconomic Pathway SSP5-8.5 (*ssp585*; henceforth referred to as *forced* in the rest of the paper), along with the historical simulations (henceforth referred to as *control*), for the assessment of the impact of anthropogenic climate change. SSP5-8.5 represents the upper boundary of the range of scenarios described in the literature, and it is an updated version of the CMIP5 RCP 8.5 scenario; by the end of the year 2100, it has an additional radiative forcing of 8.5 $Wm^{-2}$. We use the first ensemble member (*r1i1p1f1*) for each model. The models and data used are listed in Table 1. Prior to the calculations, data from each model was interpolated onto a common $2.5° \times 2.5°$ grid. The calculations presented here are performed over the years 1980-2000 for the control set and 2080-2100 for the forced set. Flux calculations are performed using daily data. Other results presented use computations with monthly data or by downsampling from daily data. Statistical significance of the changes is tested using a two-tailed Student's t-test for the difference between two independent sample means (Wilks, 2011).

#### 2.1.3 Aquaplanet simulation

We also utilize aquaplanet simulations using the Community Atmosphere Model (CAM6), the atmospheric component of the Community Earth System Model (CESM v2.1.3). We have used the finite-volume dynamical core with a horizontal resolution of $1.9°$ latitude x $2.5°$ longitude and 32 vertical levels. The control simulation is forced using the SST distribution used by Wu and Shaw (2016) to simulate the large-scale monsoonal flow, along with perpetual July insolation. The forced simulations are performed by increasing the $CO_2$ concentration to 4-times of its reference value and a uniform SST increase of 4K (Webb et al., 2017). All the other parameters are kept at their default values. Each simulation is run for three years, with the first two years discarded as spin-up. The results of these simulations are presented in Section 4.2.



**Table 1.** List of models in the *ssp585* and *historical* simulation ensembles used in this study. For all models, the variables used were daily means of *ua*, *va*, along with monthly mean *ta*, *zg* and *tos*. The variable names are as they appear in the IPCC nomenclature.

| Model | ua | va | zg | ta | tos | Model | ua | va | zg | ta | tos |
|---|---|---|---|---|---|---|---|---|---|---|---|
| ACCESS-CM2 | ✓ | ✓ | ✓ | ✓ | ✓ | BCC-CSM2-MR | ✓ | ✓ | ✓ | ✓ | ✓ |
| CESM2-WACCM | ✓ | ✓ | ✓ | ✓ | ✓ | CMCC-CM2-SR5 | ✓ | ✓ | ✓ | ✓ | ✓ |
| CMCC-ESM2 | ✓ | ✓ | ✓ | ✓ | ✓ | CanESM5 | ✓ | ✓ | ✓ | ✓ | ✓ |
| EC-Earth3 | ✓ | ✓ | ✓ | ✓ | ✓ | EC-Earth3-CC | ✓ | ✓ | ✓ | ✓ | ✓ |
| EC-Earth3-Veg | ✓ | ✓ | ✓ | ✓ | ✓ | EC-Earth3-Veg-LR | ✓ | ✓ | ✓ | ✓ | ✓ |
| GFDL-CM4 | ✓ | ✓ | ✓ | ✓ | ✓ | IITM-ESM | ✓ | ✓ | ✓ | ✓ | ✓ |
| INM-CM4-8 | ✓ | ✓ | ✓ | ✓ | ✓ | INM-CM5-0 | ✓ | ✓ | ✓ | ✓ | ✓ |
| IPSL-CM6A-LR | ✓ | ✓ | ✓ | ✓ | ✓ | KACE-1-0-G | ✓ | ✓ | ✓ | ✓ | ✓ |
| MIROC6 | ✓ | ✓ | ✓ | ✓ | ✓ | MPI-ESM1-2-HR | ✓ | ✓ | ✓ | ✓ | ✓ |
| MPI-ESM1-2-LR | ✓ | ✓ | ✓ | ✓ | ✓ | MRI-ESM2-0 | ✓ | ✓ | ✓ | ✓ | ✓ |
| NorESM2-LM | ✓ | ✓ | ✓ | ✓ | ✓ | NorESM2-MM | ✓ | ✓ | ✓ | ✓ | ✓ |
| TaiESM1 | ✓ | ✓ | ✓ | ✓ | ✓ | | | | | | |

## 2.2 Diagnostics

### 2.2.1 Zonal momentum budget

The zonally averaged zonal momentum equation reads (Dima et al., 2005; Kraucunas and Hartmann, 2005),

$$\frac{\partial [u]}{\partial t} = [v]\left(f - \frac{1}{\cos\phi}\frac{\partial [u]\cos\phi}{\partial y}\right) - \frac{1}{\cos^2\phi}\frac{\partial [u^*v^*]\cos^2\phi}{\partial y} - [\omega]\frac{\partial [u]}{\partial p} - \frac{\partial [u^*\omega^*]}{\partial p} + [\overline{X}] \tag{1}$$

The notation above is standard. Square braces denote a zonal mean, and asterisks denote a deviation from this mean. The first term on the right is the mean meridional advection, the second is the meridional eddy momentum flux convergence, and the third and fourth terms are the mean vertical advection and vertical eddy flux convergence. The last term is a residual and accounts for all sub-grid-scale processes. For Day of Year variations in the zonal momentum budget, daily estimates of each term are calculated, and then these are averaged over the respective days across all the years on record.

### 2.2.2 Helmholtz decomposition

For part of our analysis, we make use of a rotational-divergent partition of the flow. Treating the horizontal wind field on each pressure level as a two-dimensional vector field, we split it into rotational and divergent components via a Helmholtz decomposition. The horizontal velocity field is expressed as,

$$\boldsymbol{v} = \nabla\chi - \boldsymbol{k}\times\nabla\psi \tag{2}$$



$\chi$ is obtained from the inverse Laplacian of the divergence field while $\psi$ is obtained by inverting the vorticity field. The first term on the RHS describes the divergent component, while the second describes the rotational component. They will be denoted by the subscripts $d$ and $r$, respectively. Splitting the above equation into its vector components, the zonal and meridional winds can be written as

$$u = \partial_x \chi + \partial_y \psi = u_d + u_r$$

$$v = \partial_y \chi - \partial_x \psi = v_d + v_r$$

### 2.2.3   Wave activity flux

The propagation characteristics of Rossby waves activity are explored through the wave activity flux diagnostic of Plumb (1985) as it appeared in their Equation 5.7. The wave activity flux $\boldsymbol{F_s}$ takes the form

$$\boldsymbol{F_s} = p\cos\phi \begin{bmatrix} \frac{1}{2a^2\cos\phi}\left(\left(\frac{\partial\psi'}{\partial\lambda}\right)^2 - \psi'\frac{\partial^2\psi'}{\partial\lambda^2}\right) \\ \frac{1}{2a^2\cos\phi}\left(\frac{\partial\psi'}{\partial\lambda}\frac{\partial\psi'}{\partial\phi} - \psi'\frac{\partial^2\psi'}{\partial\lambda\partial\phi}\right) \end{bmatrix} \tag{3}$$

Here, primed quantities denote deviations from a zonal mean, and $\psi$ is the horizontal streamfunction estimated by inverting the vorticity field. When WKB theory is valid, i.e., when the background flow varies slowly, $\boldsymbol{F}$ ($\approx \boldsymbol{c_g}\mathcal{A}$) is the advective flux of wave activity ($\mathcal{A}$) by the group velocity ($\boldsymbol{c_g}$). For frictionless and adiabatic flow, the wave activity follows the conservation relation,

$$\frac{\partial\mathcal{A}}{\partial t} + \boldsymbol{\nabla}\cdot\boldsymbol{F_s} = 0,$$

The divergence and convergence of $\boldsymbol{F_s}$ indicate sources and sinks of wave activity flux. Further, in the zonal mean, the three-dimensional wave activity flux of Plumb (1985) reduces to the Eliassen-Palm flux (Eliassen and Palm, 1961).

## 3   Present-day picture in reanalysis

### 3.1   Zonal balance of momentum in the deep upper tropics

The annual cycle of all terms in the zonal mean zonal momentum equation (Equation 1) for the deep upper tropics is presented in Figure 1. Consistent with previous studies that focused on yearly or seasonal means (Lee, 1999; Dima et al., 2005), the horizontal eddy and mean terms lead the momentum balance of the tropical upper troposphere. Compared to the rest of the

year, these terms are particularly enhanced during the Asian summer monsoon season (June through September, Day of Year 152-273 in Figure 1) and are much smaller during the equinoctial seasons.



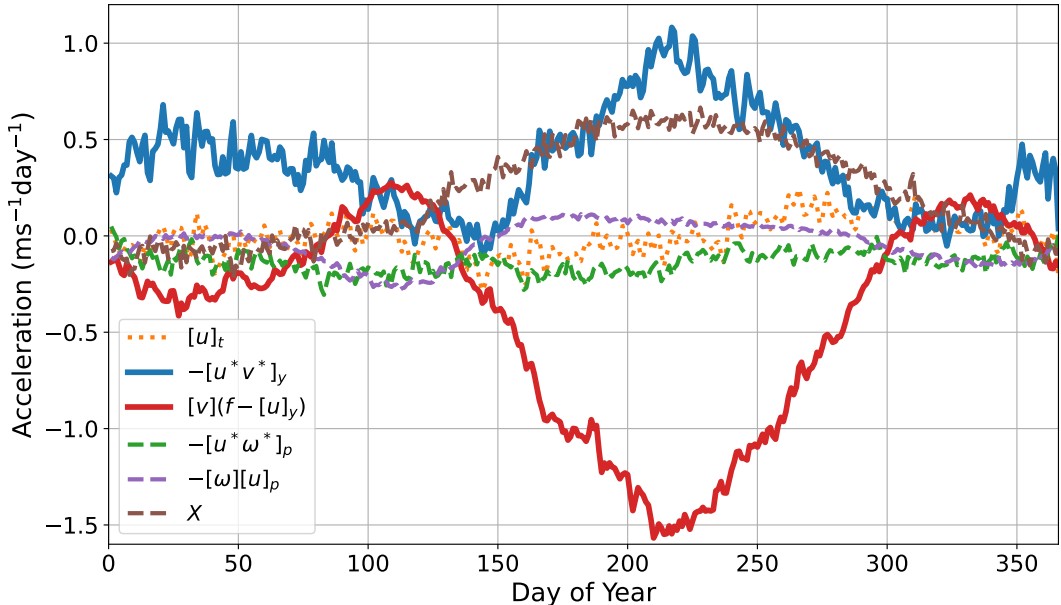

**Figure 1.** Climatological Day of Year variation of each term of the zonally averaged zonal momentum budget, Equation 1, averaged over 150-300 mbar, $\pm 5°$ of the equator.

The two dominant terms exhibit a surprising degree of anti-correlation over the course of the entire year. Dima et al. (2005) suggested that this tendency arises because the zonally averaged tropical rain bands and eddy forcing occur in the same latitudinal belt. The zonally averaged heating generates a meridional overturning circulation and a resultant upper-tropospheric

mass flux divergence, while the eddy forcing results in a convergence of eddy momentum flux into the source region. This line of reasoning is also supported by idealized modeling studies (Kraucunas and Hartmann, 2005, 2007). In fact, Kraucunas and Hartmann (2007) demonstrate that the eddy momentum flux is amplified by the presence of a mean meridional flow and is directed opposite to it. Further, Zurita-Gotor (2019) showed that the tropical eddy momentum flux is largely made up of correlations between zonally anomalous divergent wind and the climatological stationary Rossby gyres. Essentially, a

large cross-equatorial Hadley Cell flow advects vorticity across hemispheres, forcing a dipolar vortex structure straddling the equator, which results in the observed eddy momentum flux. Thus, the eddy momentum flux is always anti-correlated with the mean flow response.

The residual ($X$ in Figure 1) attains a large non-negligible value during the summer ($\sim 0.5$ ms$^{-1}$day$^{-1}$) and is much stronger than the other remaining terms. This may be due to intense convective momentum transport that is known to occur over

the Indian Ocean - Maritime Continent region during this season (Lin et al., 2008). However, Lin et al. (2008) model the convective momentum transport as a vertical convergence of zonal eddy momentum flux, which is an explicit forcing term in our momentum budget equation (Equation 1; green dashed curve in Figure 1). Thus, the budget residual ($X$) may be due to other unresolved sub-grid scale processes like gravity wave drag or due to biases in reanalysis (Carr and Bretherton, 2001).





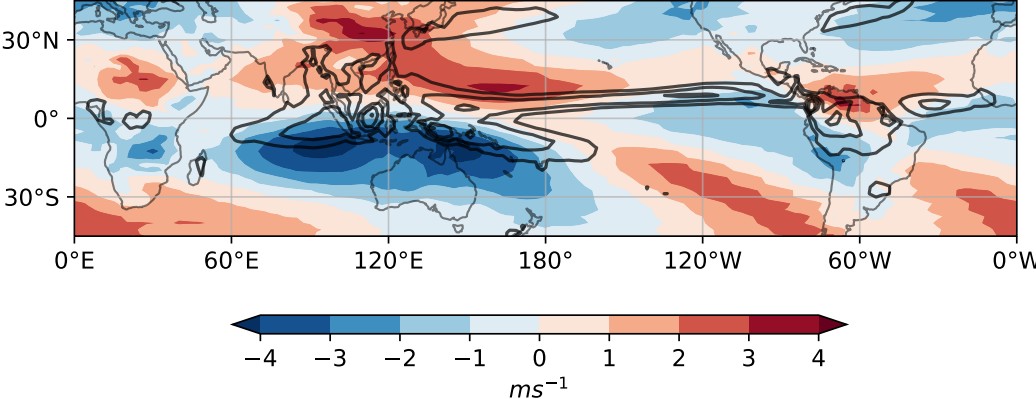

**Figure 2.** Spatial map of the annual mean difference between upper (150 mbar) and lower level (925 mbar) divergent meridional winds (colors), along with time mean precipitation (contours). The contours of precipitation are 5, 7, 9, and 11 mm/day.

The residual term weakens after the monsoon and becomes comparable in magnitude to the other terms that remain small

throughout the year. Given the systematic dominance of the mean meridional advection and eddy momentum flux convergence throughout the year, the discussion that follows will be focused entirely on these two terms.

### 3.2 Longitudinally varying contributions that make up the zonal mean

There has been increasing attention recently on the longitudinally limited portions that make up the zonal mean circulation (Raiter et al., 2020; Hoskins and Yang, 2023). In particular, it has been noted that these longitudinally restricted circulations

exhibit variability on interannual timescales (Zhang and Wang, 2013; Sun et al., 2019; Galanti et al., 2022), and may respond differently to greenhouse gas-induced climate change (Staten et al., 2019). These localized north-south overturning circulations are made up of the divergent component of the meridional wind (Zhang and Wang, 2013; Schwendike et al., 2014); the rotational contribution goes to zero when integrated over a latitude circle because it arises from a zonal gradient of the horizontal streamfunction (Equation 2). Further, these divergent motions are intimately connected to the local distribution of precipitation.

With the consideration that eddy momentum fluxes are fundamentally linked to the mean meridional flow (see Section 3.1), we construct a spatial map of the annual mean divergent meridional wind to explore the spatially varying contributions that constitute the robust zonal mean balance of momentum.

Figure 2 shows the annual mean difference between the upper and lower tropospheric divergent meridional wind along with contours of annual mean precipitation. Thus, Figure 2 emphasizes the divergent upper-level motions linked with the monsoonal

heating. The largest amount of precipitation is concentrated over the Indian Ocean - Maritime Continent region, collocated with an equatorially symmetric divergent outflow. This structure stretches across more than $120°$ in the zonal direction and more than $30°$ on either side of the equator. A similar, but smaller, equatorially symmetric structure is visible over the African continent. In the East Pacific and West Atlantic, the precipitation is zonally elongated and off-equatorial, while that over South



America has an equatorial peak over Colombia and north-western Brazil; the divergent circulation in this region is largely
hemispherically asymmetric. Further, the meridional circulation in the tropical East Pacific is observed to be shallow (Zhang
et al., 2004) and is quite different from the canonical picture of the zonal mean hemispherically symmetric Hadley circulation
(Schneider, 2006). Based on this preliminary scrutiny, we subjectively partition the tropics into two regions. These are the
Asia-Africa region (30W - 150W; abbreviated as Af-A) and the Central Pacific - West Atlantic (150W - 30W; abbreviated as
CP-WA) region.

### 3.2.1 Zonally and seasonally sensitive character of the eddies

Considering these two longitudinal sectors, Figure 3a shows their contributions to each of the leading terms of Equation 1.
For reference, the zonally averaged eddy and mean terms have been plotted again in thick blue and red curves, respectively.
Note that, in Figure 3, all eddy terms are plotted in blue and mean terms are in red. Interestingly, the climatological *DayofYear*
variation of the zonal mean eddy fluxes is determined by an internal compensation between eddies from the Af-A and CP-WA
sectors. While eddies in the Af-A (dotted blue curve in Figure 3a) tend to accelerate the flow for most of the year, eddies in the
CP-WA (dashed blue curve) have a seasonally sensitive character. They tend to decelerate the flow during the winter season but
provide a weak acceleration during summer. In the zonal mean, the decelerating tendency from CP-WA offsets a large portion
of the accelerating tendency from Af-A to give a subdued maximum of $\sim 0.5\text{m-s}^{-1}\text{-day}^{-1}$ during winter.

A rotational-divergent decomposition brings forth the detailed nature of these eddy fluxes (see Section 2.2.2; blue curves in
Figure 3b,c, and d). As shown by Zurita-Gotor (2019), the divergent component ($u_r^* v_d^*$, dash-dotted line in Figure 3b) dominates
the zonal mean eddy momentum flux convergence throughout the year. It is largely attributed to the action of tropical stationary
Rossby waves, forced by thermally divergent motions, that shift in tune with the large-scale monsoon flow. In contrast, the
rotational component ($u_r^* v_r^*$, dotted line in Figure 3b) provides a negative momentum forcing throughout the year. Typically,
the rotational component arises from eddy vorticity fluxes associated with extratropical waves (Zurita-Gotor, 2019). These
waves converge momentum into the source region, accelerating the midlatitude eddy-driven jet but diverge momentum away
from the tropics where they break and decelerate the flow (Vallis, 2017).

Figure 3c shows the rotational-divergent partition for the Af-A sector. Here too, the full eddy term follows the divergent
term ($u_r^* v_d^*$) very closely. Again, the rotational term ($u_r^* v_r^*$) provides the familiar negative momentum forcing. However, its
contribution goes to zero during the peak monsoon months; this is because extratropical waves are unable to propagate into
the deep tropics due to the strong upper-level easterlies associated with the Asian summer monsoon. In contrast, the nature of
the fluxes over the CP-WA sector is quite different (Figure 3d). The full-eddy term switches between rotational and divergent
tendencies as the seasons change from winter to summer, and vice-versa. Further, the divergent term is weak during winter but
attains a moderately large magnitude during summer. This suggests that extratropical waves are able to make their way into the
deep tropics in the CP-WA region during winter and make an important contribution to the tropical momentum balance.

To better understand the eddy momentum budget, the spatial features associated with the seasonal variation of the eddy fluxes
are shown in Figure 4. The internal compensation between the eddy fluxes from Af-A and CP-WA during winter, noted above







**Figure 3.** Panel (a) shows the climatological *Day of Year* variation of the horizontal eddy and mean terms of the upper-level momentum balance computed over the global longitudes and over the Af-A and CP-WA sectors. The eddy momentum flux convergence is presented in blue, and the mean meridional advection is presented in red. The eddy and mean momentum terms computed over the Af-A and CP-WA regions are shown in dotted and dash-dotted lines, respectively. The eddy and mean fluxes shown in thick blue and red lines are the same quantities that were shown in Figure 1, shown here again for reference. The quantities in panels (b), (c), and (d) show the rotational-divergent partition of the eddy and mean terms computed over the global longitudes, Af-A, and CP-WA sectors, respectively. The divergent component ($u_r v_d$) is shown in dash-dotted lines, while the rotational component ($u_r v_r$) is shown in dotted lines. The $u_d v_d$ and $u_d v_r$ fluxes were found to be small and have not been considered. All quantities are averaged over 150-300 mbar, and $\pm 5°$ about the equator. A 20-day low-pass filter is applied to the quantities in panels b, c, and d prior to presentation. In all panels, the *x*-axis is Day of Year and the *y*-axis is Acceleration ($\mathrm{ms^{-1} day^{-1}}$), as in Figure 1. Legend marked in panel (b) also applies to panels (c) and (d).



(see Figure 3a), arises due to the features highlighted by the black boxes in Figure 4a. The eddy momentum flux convergence in the 120E box is due to the contribution linked to the two off-equatorial anti-cyclonic Rossby gyres straddling the equator around the Maritime Continent region (Figure 4b). This flux is largely divergent in nature, as noted in Figure 3c. In contrast,

the deceleration in the East Pacific over the 120W box is due to the rotational component associated with the convergence of extratropical wave activity (Figure 3d). The Plumb flux (arrows in Figure 4b) shows the propagation of a quasi-stationary Rossby wave from the SH Pacific Ocean, across the tropics into the NH. This wave is apparently generated by deep convection in the Maritime Continent region and propagates towards the East Pacific after equatorward reflection from SH high latitudes (Goyal et al., 2021). Similar propagation characteristics have also been observed by other studies (see Figure 6 of Barnes and

Hartmann, 2012, and Figure 1 of Caballero and Anderson (2009)). While the wave activity flux in (arrows in Figure 4b) captures stationary contributions, the negative eddy momentum forcing in the 120W box is due to both stationary and transient Rossby waves. Large amplitude Rossby waves, generated in the extratropics of the NH, undergo wave breaking in the westerly duct region and intrude high PV air into the deep tropics (Waugh and Polvani, 2000; Waugh and Funatsu, 2003). Further, transient Rossby waves propagating along the SH subtropical jet can also be refracted towards the westerly duct over the central Pacific

(Matthews, 2012). The relative contributions by stationary and transient eddies will be discussed further in Section 4.3.

During the boreal summer, the net eddy fluxes in both zones are accelerating in nature. For Af-A, the dominant feature responsible for the eddy flux convergence in the deep tropics is the upper-tropospheric return flow of the Asian summer monsoon (see Dima et al., 2005; Hoskins et al., 2020, and Figure 4d) — highlighted by the box in Figure 4c. The climatological stationary Rossby gyres are now associated with the intense convection of the Asian summer monsoon and converge eddy momentum

flux into the tropics near 60E. The large-scale circulation structure is very similar to an off-equatorial Gill response (Gill, 1980; Dima et al., 2005). In stark contrast to winter, the characteristic westerly duct over the East Pacific has been replaced by anomalous weak easterlies (Figure 4c). As a result, there is no cross-equatorial propagation of extratropical Rossby waves or the related eddy momentum flux divergence in the eastern tropical Pacific. However, this region experiences a weak acceleration during summer ($\sim 0.5$m-s$^{-1}$-day$^{-1}$ in magnitude). This weak eddy acceleration is possibly related to the circulation driven

by a localized heat source (the summer inter-tropical convergence zone) (Suarez and Duffy, 1992; Kraucunas and Hartmann, 2005).

Thus, over the course of the seasonal cycle, we observe that the eddies in the Asia-Africa sector exert a positive momentum forcing on the zonal mean flow. However, those in the CP-WA sector exhibit a seasonally sensitive character, changing between negative and positive momentum forcing of the zonal mean flow as the seasons change from winter to summer. Further, the

differences in the two regions as well as changes through different seasons are due to the equatorial and extra-tropical nature of eddies that make their way into the deep tropics.

### 3.2.2 Robust nature of the mean meridional flux

A striking feature of Figure 3a is the high degree of similarity of the mean meridional fluxes when computed over the entire globe or either of the individual sectors (all red curves). This happens because the $u_r$-$v_d$ contribution is either the leading





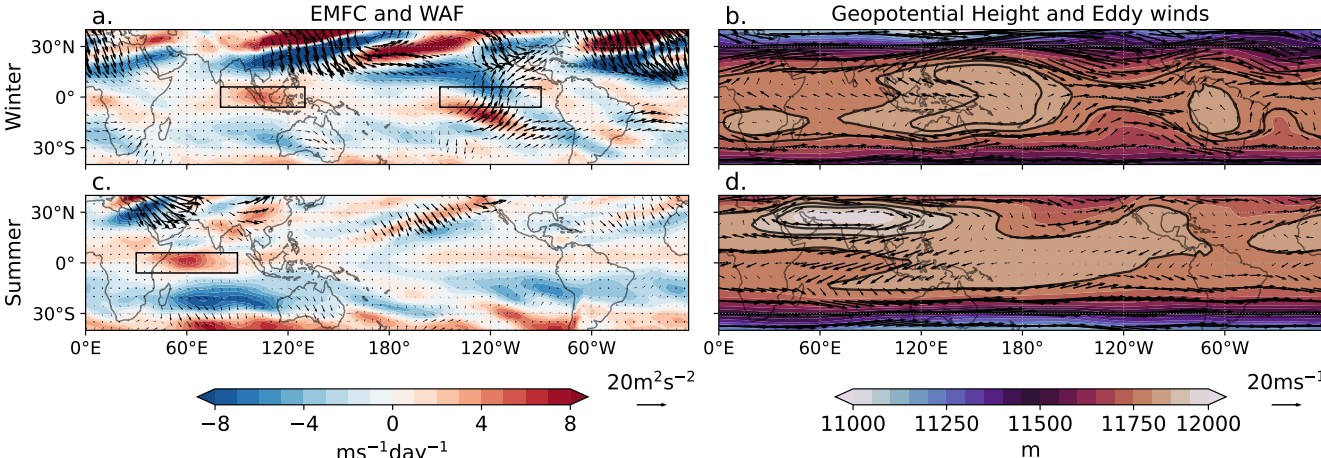

**Figure 4.** Upper-tropospheric seasonal mean distribution of (a,c) eddy momentum flux convergence and (b,d) eddy geopotential height for boreal (a,b) winter and (c,d) summer, averaged over 150-300 mbar. Overlain are quivers of seasonally averaged (a,c) wave activity flux and (b,d) eddy horizontal velocity. The wave activity flux is computed using Equation 3 (see also Plumb, 1985; Caballero and Anderson, 2009).

influence or the only factor that determines the mean meridional advection of momentum over each sector (solid and dashed red curves of Figure 3b,c, and d). On splitting Equation 2 into its zonal and meridional wind components, it can be readily seen that $v_r$ and $u_d$ are zonal gradients of a stream function and velocity potential, respectively (see Section 2.2.2), and go to zero when integrated over all longitudes. With this consideration, the mean meridional momentum advection term can be written as $[v_d](f - \partial_y[u_r])$, which is the advection of zonal mean absolute vorticity by the zonal mean divergent meridional wind

from the summer hemisphere into the winter hemisphere. Considering the annual cycle of $v_d$ and $-\partial_y u_r$, averaged over the global longitudes and the individual sectors Af-A and CP-WA (Figure S1), we find that these quantities have a similar character irrespective of their domain. Thus, this theme of cross-equatorial transport of vorticity by the thermally direct circulation is consistent across all domains considered in this study (Figure S1).

As long as $u_r$-$v_d$ or tropical features dominate the eddy and mean fluxes, they should oppose each other in strength and

symmetry, as seen in the substantial anti-correlation between the zonal mean eddy and mean fluxes (see Section 3.1). This anti-correlation appears to be valid in the Af-A, with seasonally changing peaks of similar magnitude in both the eddy and mean fluxes (panels a and c of Figure 3). However, the CP-WA sector is an exception; the eddy flux is much weaker than the mean flow response (panels a and d of Figure 3) — indeed, this is due to the substantial contribution of the rotational extratropical eddy momentum fluxes in this region.

In all, the eddy momentum fluxes that participate in the zonal mean balance are quite diverse. The fluxes are dissimilar in winter, accelerating the zonal flow over Af-A while decelerating it over CP-WA. In summer, both these regions experience acceleration due to the general convergent tendency of the eddy fluxes. In the zonal mean, the mean meridional momentum advection balances these eddy fluxes. The magnitude of the mean term in each zone peaks in the boreal summer, despite





representing different longitudinal zones. Further, the nature of the mean meridional momentum flux is similar for both the individual regions and the zonal mean throughout the year. With these considerations, it becomes evident that the tropical momentum balance is quite delicate. It involves a three-way compensation between tropical eddy acceleration, extratropical eddy deceleration, and zonal-mean absolute vorticity advection by the divergent meridional flow. A change in any of these three components has the potential to alter the upper tropical momentum budget. Given the many changes anticipated in the tropical circulation due to global warming (see the discussion in the Introduction), it would be worthwhile to explore any influence of warming on these fluxes and whether there can be a change in the tropical momentum balance.

## 4 Momentum fluxes in a warming climate

To see if such an analysis is feasible, we begin by examining the upper-tropospheric eddy and mean momentum flux terms for the control simulation of the CMIP6 multi-model ensemble (Figure 5a). The control set qualitatively captures the mean meridional momentum flux term, globally and locally over the Af-A and CP-WA sectors (compare Figures 3a and 5a). It also captures the dissimilarity in winter-time eddy fluxes in the Af-A and CP-WA regions. While of the correct sign, some discrepancies are apparent. Firstly, the control run has a weaker eddy acceleration than that observed in the present-day reanalysis. Secondly, the mean meridional deceleration in the CP-WA sector is larger in magnitude than that noted in the reanalysis. A possible reason for these discrepancies is that the model fluxes tend to be slightly displaced than those for present-day reanalysis. Such discrepancies are possibly associated with systematic biases suffered by these models (Wang et al., 2014; Mitchell et al., 2020). Further, these discrepancies may arise from the inability of these models to simulate atmospheric stationary waves correctly (Boyle, 2006), which in turn may be due to the shortcomings of the modeled representation of stationary wave forcing mechanisms (see, for example, Held et al., 2002). Additionally, Frierson and Hwang (2012) show that the inter-model spread in the ITCZ response can be attributed to the response of extratropical clouds to external forcing. The high-latitude radiative perturbations strongly influence the ITCZ position (Kang, 2020). Many of these climate models struggle to reproduce the delicate climatological balance between the warm pool, cold tongue, and the ITCZ (see Pierrehumbert, 2000, and the references therein). Nevertheless, given the overall qualitative agreement between the control and present-day estimates, it is worthwhile to use the CMIP6 suite for studying the influence of climate change on upper-tropospheric momentum fluxes. Momentum fluxes from the forced set suggest changes in both the summer and winter seasons when compared to the control (Figures 5a and b). For example, the mean meridional term computed over the CP-WA weakens considerably. Although this anomaly is noticeable primarily over the CP-WA sector, it manifests in the globally averaged term by extension. We also notice changes in the eddy acceleration in both the CP-WA and Af-A regions during summer. Overall, the magnitude of momentum fluxes decreases due to warming.

One of the most robust responses to anthropogenic climate change is an increase in the tropopause height (Lorenz and DeWeaver, 2007; Vallis et al., 2015). As eddy fluxes of momentum tend to be concentrated in the upper troposphere (Ait-Chaalal and Schneider, 2015), there arises a possibility wherein fluxes undergo an upward displacement along with the tropopause. Further, idealised and full-complexity simulations suggest that the Hadley circulation is expected to undergo a





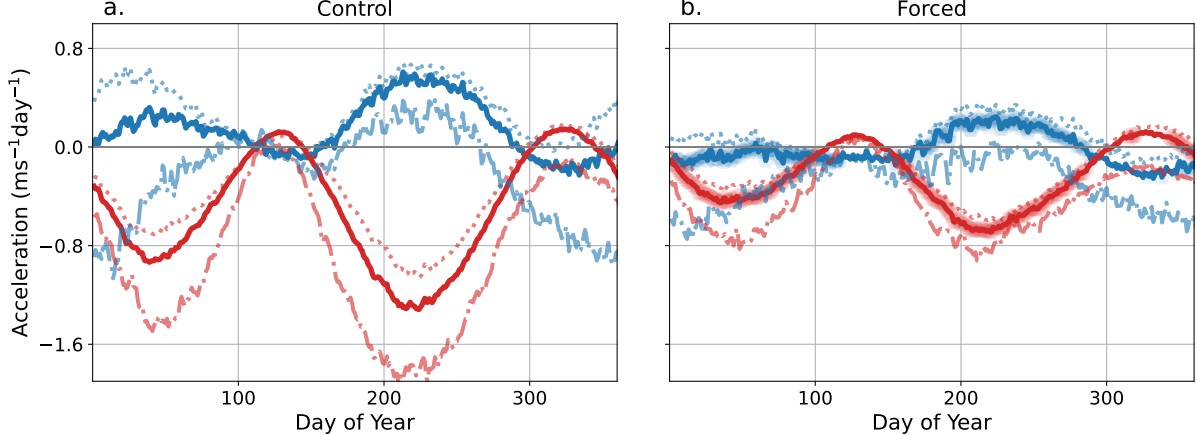

**Figure 5.** Same as Figure 3a, except for CMIP6 (left) control and (right) forced sets. The blurring of the curves, in panel b, indicates days where the full eddy and mean terms related to the forced simulations are statistically different from those for control at the 95% level of significance.

weakening and poleward expansion due to warming (Lu et al., 2007; Frierson et al., 2007). Thus, rather than focusing on a particular set of pressure levels, a view of the fluxes with height and latitude is more appropriate; this is shown in Figures 6 and 7 for summer and winter seasons, respectively. As with the upper layer averages (Figures 3a,e), the latitude-height computations

for the control set bears a resemblance to reanalysis (see Dima et al., 2005, and Figure S2). However, the magnitudes of the terms are smaller in the control set than those for the reanalysis.

## 4.1 Changes in the latitude-height plane

In response to anthropogenic warming, the near-equatorial eddy acceleration shifts upward and is concentrated near the 100 mbar level in the forced set (comparing Figures 6a and g). This is consistent with expectations of an increase in tropical

tropopause height and an upward shift in the circulation with warming (Singh and O'Gorman, 2012). There is a clear and significant increase in the magnitude of the zonal mean eddy fluxes during summer (compare Figures 6a and g), largely due to the increase over the Af-A sector that can be observed in both the upper troposphere and lower stratosphere (compare Figures 6b and h). This large increase of the eddy momentum flux over the Af-A sector is possibly related to changes in the Asian summer monsoon anti-cyclone whose signature extends up to 10 mbar in present-day observations (Krishnamurti et al., 2008);

this aspect will be explored further in Section 4.2. A similar increment is visible over this region in the winter season as well (Figures 7b and h).

In comparison, the response of the CP-WA sector is different in the summer and winter seasons, similar to its seasonally sensitive behavior observed in the present-day climate (see the discussion in Section 3.1). The tropical eddy momentum flux convergence is spread over a deeper vertical layer in summer, extending up to the tropical tropopause (Figures 6c and i).





**Figure 6.** Seasonally and zonally averaged multi-model mean vertical structure of (a-c & g-i) eddy momentum flux convergence and (d-f & j-l) mean meridional momentum flux convergence for the CMIP6 (top half, a-f) control and (bottom half, g-l) forced simulations for summer (JJA). The quantities displayed are computed over (left column) the global longitudes, (middle) Af-A, and (right) CP-WA, respectively. Logarithmically-spaced contours have been plotted additionally for more clarity. Positive values are solid while negative values are dashed, and the zero contour is dotted. The lowest non-zero value represented by the contours is 0.1 m-s$^{-1}$-day$^{-1}$, and the subsequent contours double in magnitude. In each panel, the red curve is indicative of the multi-model mean tropopause height over the corresponding region (see Reichler et al., 2003). Hatching denotes statistically significant changes at the 95% level by a two-tailed t-test.

Whereas, during winter, the signature of the eddies weakens on either side of the equator (Figures 7c and i). This is possibly







**Figure 7.** Same as Figure 6 except for seasonal mean over winter (DJF).

related to changes in the equatorward propagation of extratropical wave activity through the westerly duct (see discussion in Section 3.1) and will be explored further in Section 4.3.

Further insight regarding changes in the vertical structure of the eddy fluxes is gained by splitting the eddy momentum flux into stationary and transient contributions (Figure 8; Lee, 1999; Peixoto and Oort, 1992). We note that the present-day tropical momentum balance is largely dominated by the stationary eddy component (compare panels a and d of Figure 8). Although the peak values are similar, the stationary contribution is spread over a larger area than the transients, whose influence is confined to a small region in the deep tropics near 150 mbar. Further, the multi-model ensemble means of these quantities compare



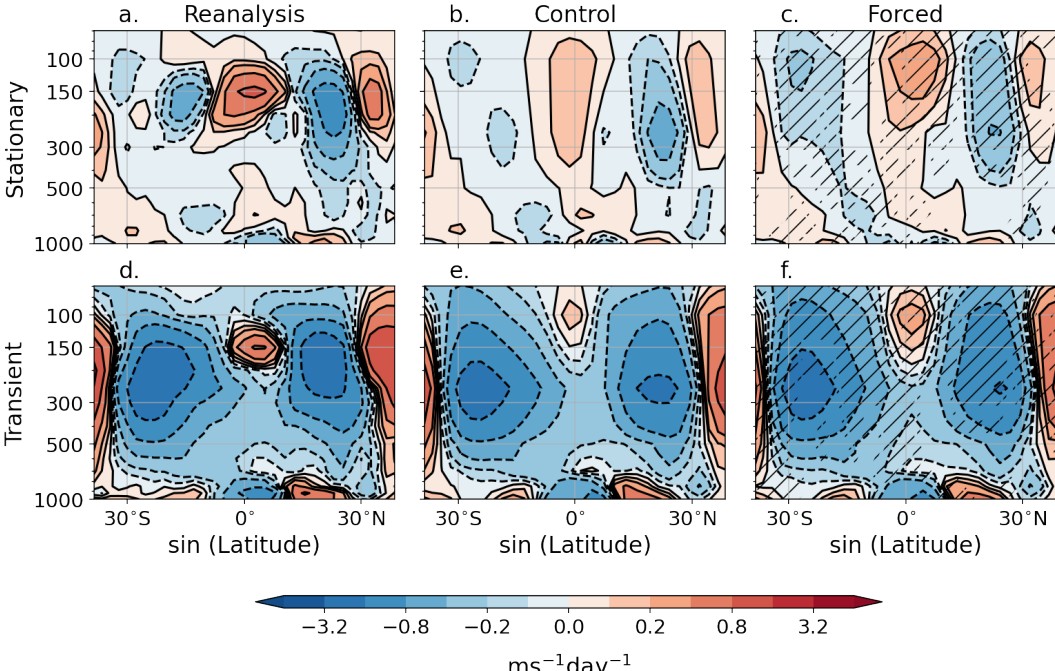

**Figure 8.** Temporally and zonally averaged estimates of eddy momentum flux convergence due to (a,b,c) stationary eddies and (d,e,f) transient eddies (Lee, 1999) for (a,d) reanalysis, (b,e) CMIP6 control and (c,f) forced set. Hatching denotes changes that are statistically significant at the 95% level by a two-tailed t-test.

poorly with reanalysis, both in magnitude and position (compare panels a,b, and d,e in Figure 8). The stationary eddies in the control set appear to be spread across the entire troposphere and the stratosphere as opposed to their expected position near
the tropopause (Ait-Chaalal and Schneider, 2015). The transient eddies appear to peak at 100 mbar rather than 150 mbar, as in the reanalysis. In both cases, the maxima in the tropical eddy momentum flux convergence is $\sim$0.2m-s$^{-1}$-day$^{-1}$, which is far less than the $\sim$1.6m-s$^{-1}$-day$^{-1}$ peak in the reanalysis. However, in response to warming, both these quantities exhibit an increase in their magnitude (compare panels b,c, and e,f in Figure 8). Similar to the changes observed in the Af-A sector, the stationary eddy momentum flux appears to shift upwards and shows a large increase in magnitude with warming. The increase
in transient eddy momentum flux in the tropics may be due to increased MJO amplitude with warming, as seen in externally forced idealized aquaplanet (Lee, 1999; Arnold et al., 2013) and super-parameterized Earth system model simulations (Arnold et al., 2015). However, similar diagnosis in IPCC-class CMIP models has remained elusive (Maloney et al., 2019).

The mean meridional flux undergoes a weakening and upward spread during both seasons (compare panels d, and j in Figures 6 7). Under the influence of warming, the peak in the flux over Af-A shifts close to the tropopause in summer (compare e,k
in Figure 6), while that over the CP-WA decreases in strength (compare f,l in Figure 6)). The mean flux undergoes a general weakening across all sectors during winter (compare e,k in Figure 7 and f,l in Figure 7). These changes in the mean meridional



flux are anticipated by theories that posit a wider and weaker Hadley cell in a warmer climate (Lu et al., 2007; Frierson et al., 2007; Levine and Schneider, 2011; Vallis et al., 2015). Further, the response of the Hadley circulation to greenhouse gas radiative forcing is spatially complex and involves intricate compensations between varied responses of the mean flow to external radiative forcing (Kim et al., 2022).

On the other hand, the changing nature of eddy fluxes, their regional dependence, and their signature in the upper troposphere and lower stratosphere is intriguing and, to the best of our knowledge, has not received much attention. We now probe the cause of these changes, with a particular focus on Af-A during summer and CP-WA during winter.

## 4.2  Changes in the Asia-Africa region

The most prominent feature of the tropical atmospheric circulation during the NH summer is the Asian summer monsoon (Hoskins and Yang, 2021). As noted in Section 3.1, the convergence of eddy momentum in the Af-A region is due to the outflow of the Asian summer monsoon anti-cyclone. Comparing the $15 \times 10^6$ m$^2$s$^{-1}$ streamfunction contour of this vortex in the control and forced runs (panels a and c in Figure 9), we find that the intensity of the circulation weakens in response to warming. Specifically, the spatial extent of this contour decreases from being spread across a broad area spanning the core zone of the Indian summer monsoon along with parts of China, Pakistan, Afghanistan, and Iran in the control set to being confined to a much smaller region over China and north-east India in the forced set. This is consistent with the projections made using previous generations of CMIP models (Ueda et al., 2006; Sooraj et al., 2015) and with high-resolution GCMs (Hsu et al., 2012). The weakening of the monsoon circulation in the upper troposphere is the net result of two competing effects of climate change; the weakening effect due to smaller sea surface temperature gradients dominates the strengthening influence of increased upward mass flux caused by the uplift of the tropopause (Levine and Boos, 2016). Further, the eddy circulations appear more zonal in the forced set. This is consistent with the simulations of Joseph et al. (2004, see also Wills et al. (2019)). Thus, the weaker eddy circulations in the upper troposphere explain the decrease in the tropical eddy momentum flux in summer (compare blue curves in Figures 3e and f).

On the other hand, Figures 9b,d suggest a strengthening of the subtropical stationary waves and the associated eddy fluxes in the lower stratosphere (100 mbar). The increased magnitude of zonally asymmetric anomalies appears to be in close association with projections of a stronger zonal mean Brewer-Dobson circulation (BDC; Butchart, 2014). The westward drag induced by upward propagating planetary-scale Rossby waves is balanced by the eastward Coriolis force on poleward-flowing air pumped upwards in the tropics, through the *downward control* mechanism (Butchart, 2014). The stronger BDC is typically attributed to increased planetary wave activity flux (McLandress and Shepherd, 2009), and the upward shift of critical layers into the lower stratosphere due to the warming (and the attendant drag due to wave breaking, Shepherd and McLandress, 2011). As more than 60% of the mean upward mass flux is due to wave-driving by resolved waves (Butchart et al., 2006; McLandress and Shepherd, 2009), a larger wave-driving of the BDC should be accompanied by a larger tropical upwelling (Butchart, 2014). However, Oberländer-Hayn et al. (2016) suggests that most of the BDC trend is associated with an upward shift of the




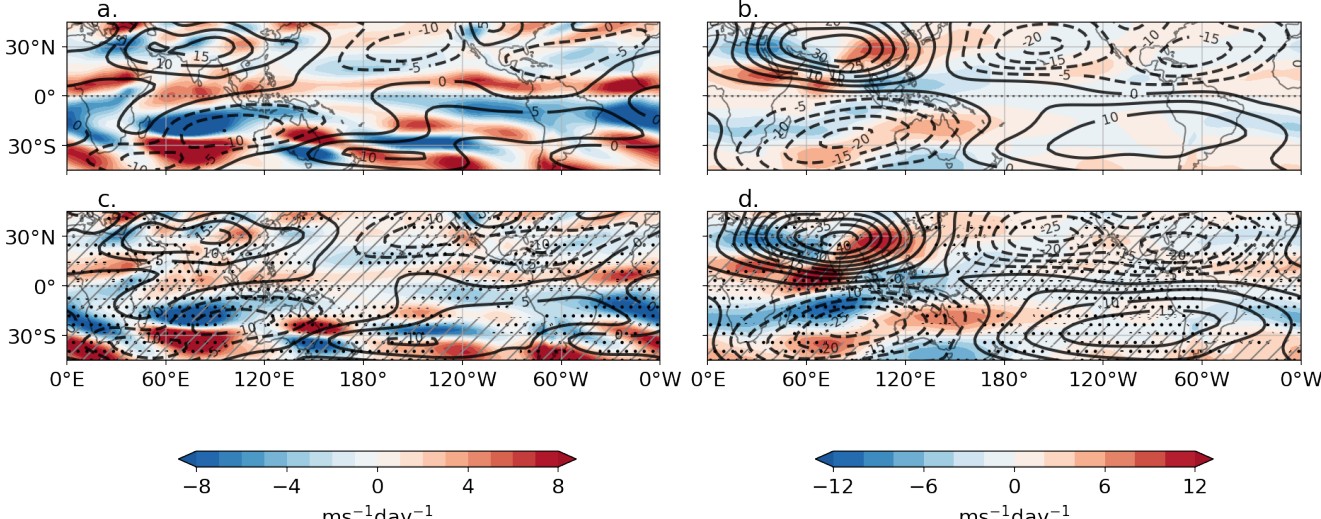

**Figure 9.** Boreal summer seasonal mean upper-tropospheric (a,c; 250 mbar) and lower-stratospheric (b,d; 100 mbar) spatial maps of eddy momentum flux convergence (colours) and eddy streamfunction (contours). The top (a,b) panels show the ensemble mean for control, while the bottom (c,d) shows that for forced simulations. Contour intervals are the same in all panels, and the units are $10^6$ m$^2$s$^{-1}$. Hatching with gray lines (stippling with black dots) denotes changes in streamfunction (eddy momentum flux convergence) that are statistically significant at the 95% level by a two-tailed t-test.

circulation. Recently, coupled climate model simulations by Menzel et al. (2023) suggest that there is no dynamical coupling between the upper tropospheric and lower stratospheric upwelling.

Irrespective of the extent of wave-driving or the cause of tropical upwelling, a larger tropical mass flux is expected with warming due to the deepening of tropical convection (Lau and Kim, 2015; Chou and Chen, 2010). Here, we propose an alternative way in which the lower stratospheric eddy fluxes are strengthened over the Asia-Africa region. The increased subtropical mass flux into the stratosphere during the NH summer, plausibly from the Indian summer monsoon (Randel and Park, 2006; Randel et al., 2010), is likely to result in a more extensive local lower-stratospheric divergence and hence a stronger rotational flow. This can be elucidated by a vorticity balance for the lower-stratospheric flow. As seen in Sardeshmukh and Held (1984), the steady-state vorticity equation for large-scale flow can be written as,

$$0 \approx -\boldsymbol{v} \cdot \nabla \zeta - \beta v - (f + \zeta) \nabla \cdot \boldsymbol{v}. \tag{4}$$

The notation in the above equation is standard; $\boldsymbol{v}$ is the horizontal velocity, $\zeta$ is the relative vorticity, $f$ is the Coriolis parameter, and $\beta$ is the meridional gradient of the Coriolis parameter. Linearizing the above equation about the zonal mean horizontal winds, and collecting the terms that are dominant in the lower stratospheric tropical region using reanalysis data (not shown), yields a dominant Sverdrup balance for the atmosphere,

$$\beta v^* + f \nabla \cdot \boldsymbol{v}^* \approx 0$$



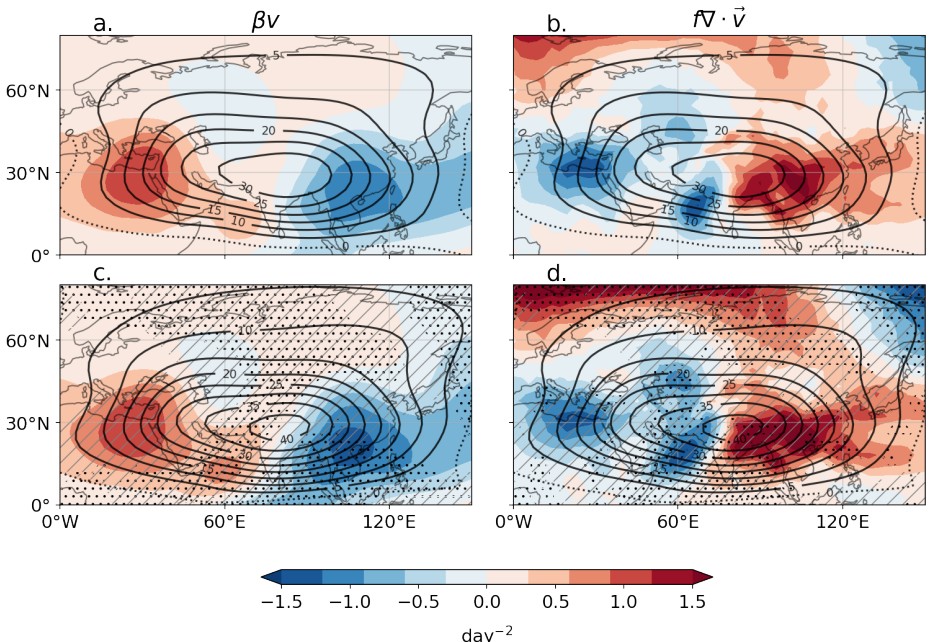

**Figure 10.** Boreal summer seasonal mean lower-stratospheric (100 mbar) dominant terms of the linearised vorticity budget (colors) and eddy streamfunction (contours) for control (top) and forced (bottom) simulations. Terms of the vorticity budget are (a,c) $-\beta v^*$, and (b,d) $-f\nabla \cdot \boldsymbol{v}^*$. Contour intervals are the same in all panels, and the units are $10^6$ m$^2$s$^{-1}$. Hatching with gray lines (stippling with black dots) denotes global warming-induced changes in the streamfunction (vorticity budget terms) that are statistically significant at the 95% level by a two-tailed t-test.

Here, $^*$ indicates a deviation from the zonal mean. In both the control and forced sets (Figure 10), the primary balance is
between the advection of planetary vorticity and the stretching term, implying a Sverdrup-like balance of the large-scale lower-stratospheric flow. Quite clearly, comparing the balance for the two scenarios suggests a higher degree of compensation between the beta and stretching terms via a larger divergence in the forced ensemble than the control set.

Given that the changes observed in the vertical structure of the Asian monsoon circulation appear to be due to relatively straightforward dynamical interactions, we now examine if they can be reproduced in the simple, bare-bones setting of an
idealized aquaplanet model (see Section 2.1.3 for details). We adopt the idea that by forcing an aquaplanet with a planetary scale wave-1 SST perturbation (see Figure S3; see also Wu and Shaw, 2016), it is possible to replicate the large-scale monsoonal flow. Figure 11a shows the latitude-height section of the horizontal streamfunction, zonally averaged over the forcing longitudes. Clearly, the model setup is able to reproduce the large-scale NH monsoon circulation with low-level cyclonic flow and upper-level anti-cyclonic flow.

Anthropogenic climate change is simulated in the model by quadrupling the reference carbon dioxide concentration and uniformly increasing the SSTs everywhere by 4K (Webb et al., 2017). Figure 11b shows the streamfunction anomaly induced



**Figure 11.** Steady-state statistics for the aquaplanet runs averaged over year 3 of the simulation. (Top; a,b) Vertical structure of eddy streamfunction ($\psi$; units: $10^6$ m$^2$s$^{-1}$) for (a) control simulation and (b) difference between forced and control simulations. (Middle; c,d) Eddy pressure velocity ($\omega$; units: hPa-day $^{-1}$) distribution for (c) control and (d) forced simulations. (Bottom; e,f) Eddy divergence field ($\nabla \cdot \boldsymbol{u}$; units: $10^6$ s$^{-1}$) distribution for (c) control and (d) forced simulations. The results in panels (a,b) are averaged over 120°E-120°W because the peak of the SST forcing lies in this longitude band. The results in panels (c,d,e,f) are all averaged over 60-100 mbar.

by the climate change forcing. The lower tropospheric cyclonic circulation strengthens, but the upper tropospheric circulation weakens; this is consistent with the paradoxical Asian summer monsoon response to warming simulated by full-complexity CMIP models (Ma and Yu, 2014). Further, consistent with the results presented so far, the model tropopause shifts upwards (compare panels a,b in Figure 11), and the lower stratospheric circulation becomes more intense with warming. As noted in




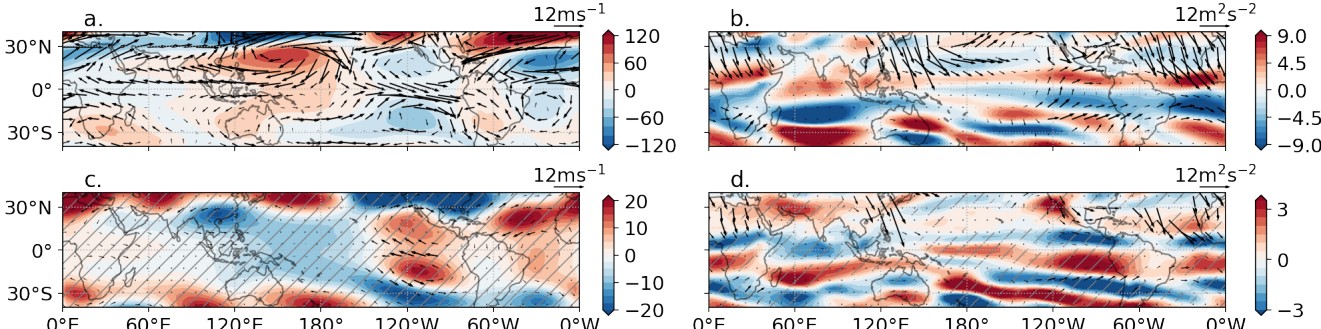

**Figure 12.** Boreal winter seasonal mean distribution of eddy geopotential height (left; units: m) and eddy momentum flux convergence (right; units: ms$^{-1}$day$^{-1}$), both for control (top) and the difference between forced and control simulations (bottom), at 250 mbar. Quivers represent eddy wind vectors (left) and wave activity flux vectors (right) computed at the same pressure level. In the climate change anomaly panels (c,d), the quivers are plotted at any point only if either $x$- or $y$- component of the respective quantity is statistically significant at 95% level at that point. Hatching denotes areas where the geopotential height (eddy momentum flux convergence) is statistically significant at the 95% level.

the CMIP6 suite, the aquaplanet simulations also suggest a larger mass flux from the troposphere to the stratosphere (Figures 11c,d) that forces a greater divergence (Figure 11e,f) in the lower stratosphere. This, in turn, leads to a stronger rotational flow in the lower stratosphere in the climate-change-forced simulation.

Indeed, deeper convection and larger troposphere-stratosphere mass flux are anticipated in warming scenarios for winter as well. However, a brief look at the lower stratospheric vorticity balances over the Af-A sector during winter suggests important contributions from non-linear terms (Sardeshmukh and Hoskins, 1985). This is confirmed by Wang and Kushner (2011), who observe that the performance of their stationary wave model in simulating the NH winter stationary waves degrades when non-linearities are neglected. Hence, the underlying physical processes are possibly more involved than the elegant Sverdrup balance seen during summer. Further, climate change projections suggest that the trend of increased tropical upwelling and stronger BDC are larger in winter (Butchart et al., 2006; Butchart, 2014). Perhaps, the stronger winter polar vortex allows stationary planetary-scale waves to propagate higher up into the stratosphere, leading to a deeper BDC through "stratospheric control" (Gerber, 2012).

It is possible that the changes in the summertime eddy flux over the CP-WA region (noted in Section 4.1, compare panels c and i of Figure 6) occur through similar dynamical interactions as those noted for the Af-A sector. This is because tropospheric vertical velocities strengthen in the East Pacific (Knutson and Manabe, 1995; Kang et al., 2023), and may lead to stronger eddy momentum flux convergence due to deeper and more intense convection. This would explain the strengthening of lower stratospheric rotational flow across all longitudes (Figure 9b,d). However, in the next section, we explicitly focus on the wintertime change that is characterized by a significant decrease in the magnitude of the eddy momentum flux in the upper troposphere (Figure 7c and i).





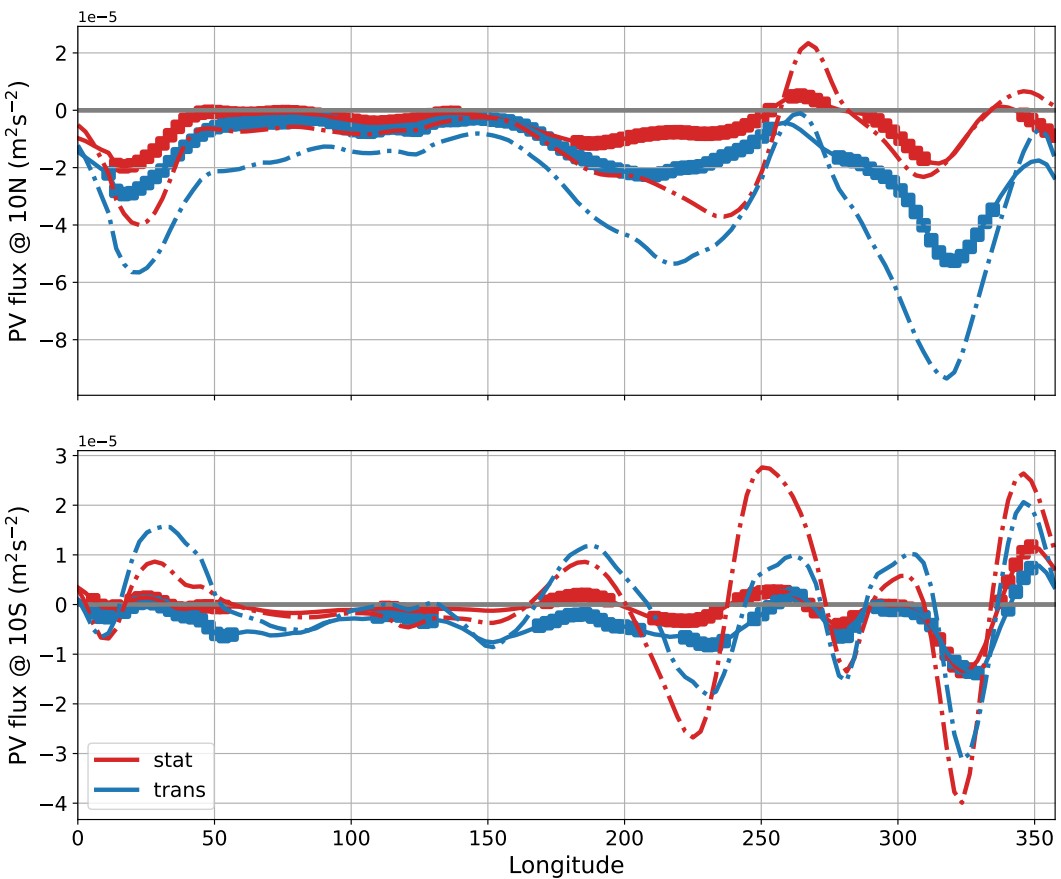

**Figure 13.** Stationary (red) and transient (blue) fluxes of PV along (top) $10°$N and (bottom) $10°$S on the 350 K isentrope averaged over winter (DJF). The solid (dash-dotted) curve represents the forced (control) set. Points marked with squares are statistically significant at the 95% level.

## 4.3 Changes in the East Pacific sector

Figure 12 captures the change that occurs in winter over the CP-WA sector. In comparison to the control set, the warming-induced changes suggest the appearance of anticyclonic anomalies straddling the equator accompanied by eddy easterlies over the East Pacific (compare panels a and c of Figure 12). This suggests that the *westerly duct* weakens with warming, and this is possibly due to the general weakening of the upper tropospheric stationary circulation pattern with warming (see Figure 1b of Wills et al., 2019). As seen for the Af-A sector during summer, these stationary features weaken due to the larger weakening effect of the decrease in horizontal thermal gradients (Levine and Boos, 2016). Locally, over the East Pacific, this appears to be due to the weaker SST gradient associated with the appearance of the warm SST anomaly simulated by CMIP models in response to warming (DiNezio et al., 2009; Xie et al., 2010).





In turn, this is accompanied by a marked decrease in the eddy momentum fluxes on either side of the equator (as seen in Section
4.1), along with a complete shut-off of the cross-equatorial propagation of wave activity over this region (Figure 12d; note that
the arrows in this panel are for the *forced* set, rather than *forced - control* as in 12c). The warming-induced change in the basic
state background winds would impact the propagation characteristics of transient eddies as well. However, the wave activity
fluxes of Figure 12 only account for the stationary waves. Since these eddies are associated with the equatorward transport of
PV across latitude circles (Ortega et al., 2018), we employ the PV flux metric on the 350K isentrope to discern changes in the
eddy activity. In the control set, the 350K surface lies close to the 200 mbar surface (Figure S4). As the tropical troposphere
warms, the 350K surface descends closer to the 300 mbar surface in the tropics. Since both these levels stay within the upper
troposphere, we consider PV fluxes on the 350K isentrope.

On a given $\theta$ surface, the PV flux is defined as (Ortega et al., 2018),

$$J = pv$$

where $p$ is the mass-weighted PV, and $v$ is the meridional wind, both in isentropic coordinates. We further decompose this
flux into stationary and transient eddy contributions, following the standard Reynolds' decomposition (Lee, 1999; Peixoto and
Oort, 1992).

Figure 13 shows the stationary and transient fluxes of PV in red and blue curves at $\pm 10°$. In the control set (dash-dotted curves),
the transient fluxes in the NH are generally equatorward and are larger in magnitude than the stationary fluxes (compare the
dashed red and blue curves in Figure 13). The transient disturbances originate in the extratropics and propagate towards the
equator (Edmon et al., 1980). The transient PV fluxes are weaker in the SH than in the NH. Consistent with the wave activity
flux pattern (see Figures 12b and 4b), we find equatorward stationary PV fluxes due to extratropical stationary waves. On either
side of the equator, the stationary flux has a northward component in the vicinity of the westerly duct. These features are in
general agreement with the discussion in Section 3.

In response to warming, there is a sharp decrease in both the stationary and transient fluxes across all longitudes. Consistent
with the wave activity flux for the forced set in Figure 12d, the northward component of the stationary flux has almost disap-
peared. The decrease in the NH transient fluxes is generally larger than the stationary fluxes. These changes in the stationary
and transient PV fluxes are consistent with the warming-induced changes in the extratropical stationary and transient eddy
momentum fluxes, particularly in the NH (see Figure 8). Further, the warming-induced changes in the wave activity fluxes may
be due to changes in the propagation characteristics due to the changes in the basic state background winds.

The warming-induced changes noted over the Central and Eastern Pacific are at odds with the historical trends in reanalysis
data, which suggest an increase in the PV intrusions over this area (Nath et al., 2017). Such trends in observations have been
attributed to the strengthening of the equatorial westerly window and, more fundamentally, the strengthening trend of the equa-
torial Pacific SST gradient. Thus, this is a manifestation of the dichotomy between observed "La Nina"-like strengthening and
simulated "El Nino"-like weakening trends of the zonal SST gradients (Lee et al., 2022). The disagreement is usually attributed



to model biases in simulating the equatorial Pacific cold tongue (Seager et al., 2019) or to multi-decadal internal climate vari-
ability obscuring the forced response (Watanabe et al., 2021). It has been suggested that models with poor representations of
ENSO variability show the largest ENSO-like climate change in the tropical Pacific (Collins, 2005; Karamperidou et al., 2017).
Further, models with larger Southern Ocean heat uptake exhibit lesser weakening of the zonal SST gradient (Kang et al., 2023).
In all, future global and regional climate projections critically depend on the forced response of the equatorial Pacific Ocean.

## 5    Summary and Discussion

The zonal mean momentum budget of the deep upper tropics is primarily known to be a balance between eddy momentum
flux convergence and mean meridional momentum advection; these terms offset each other on seasonal and annual timescales
(Lee, 1999; Dima et al., 2005). Since our motivation was to unravel the longitudinal heterogeneity that is a hallmark of our
Earth system, we have focused on the regional contributions that sum to the zonal mean. Based on the zonal distribution
of monsoonal diabatic heating, we subjectively partition the tropics into the zones of Africa-Asia (Af-A; 30W - 150W) and
the Central Pacific-West Atlantic (CP-WA; 150W - 30W). The nature of the fluxes in these geographically limited regions is
considered.

The Af-A sector provides the bulk of the eddy acceleration during the solstitial seasons. This is due to the action of the climato-
logical stationary Rossby waves, forced by thermally divergent motions. These waves are present in this region throughout the
year and shift in tune with the prevalent monsoon flow. The contribution from the CP-WA sector undergoes a semi-annual re-
versal; the eddies tend to accelerate the zonal flow in summer but exert a deceleration in winter. Analogous to the Asian sector,
divergent motions forced by the summer ITCZ and the global monsoons converge eddy fluxes of momentum into this region
during summer. However, during winter, the CP-WA sector experiences a deceleration due to the influx of extratropical wave
activity in the East Pacific. A Helmholtz decomposition of the eddy fluxes concisely captures these features. Divergent eddy
momentum flux characterizes the Af-A sector throughout the year, while the CP-WA undergoes a seasonal switch between
rotational and divergent fluxes. In comparison, the mean meridional momentum advection has a similar character in the zonal
mean and the individual sectors. Both sectors contribute cohesively toward the zonally averaged mean flow deceleration term
via the advection of absolute vorticity by divergent meridional winds. Thus, the two-way balance between zonal mean terms is
actually comprised of a seasonally sensitive and delicate three-way balance involving eddy fluxes of tropical and extratropical
origin and the mean meridional momentum flux convergence.

The expanding width of the Tropics (Collins et al., 2013; Vallis et al., 2015), changing precipitation patterns and monsoon
circulations (Ma et al., 2018; Wang et al., 2020) and weakening of the equatorial Pacific SST gradient (DiNezio et al., 2009;
Xie et al., 2010) are some of the projections for a warmer Tropical climate. These changes suggest that the impact of climate
change is going to be felt strongly, even at a regional level. Given that the tropical momentum balance is delicate on both
geographical and seasonal scales and that changes in the diabatic heating distribution associated with changing precipitation
patterns may induce changes in both the eddy and the mean flow terms, externally forced warming can potentially change the



balance of momentum and, therefore, the zonal mean zonal winds as well. We analyze these possible changes in the second part of this study using the *historical* and *ssp585* multi-model ensembles from the CMIP6 archive. Along with well-known robust responses of the circulation to externally forced warming, like the upward shift of the tropopause and a decrease in the mean meridional momentum advection term due to the weakening Hadley Cell intensity, we observe statistically significant changes
in eddy fluxes due to warming. As in the present-day climate, changes observed in the zonal mean closely resemble those in the Af-A sector. In the tropics, the eddy flux term shifts upwards along with the tropopause, decreasing the eddy momentum flux convergence in the upper troposphere but increasing it in the lower stratosphere. These changes are largely in the stationary component of the eddy flux, with a minor contribution from transient tropical eddies like the MJO.

Changes in summer are most pronounced in the Asian region and are observed in both the upper troposphere and the lower
stratosphere. The Asian monsoon anticyclone weakens in the upper troposphere and strengthens in the lower stratosphere. The upper tropospheric change is well known (Ueda et al., 2006; Sooraj et al., 2015; Hsu et al., 2012), and happens due to the larger weakening effect of smaller SST gradients with warming as opposed to the strengthening effect of warming-induced tropopause uplift (Levine and Boos, 2016). The lower stratospheric increase is larger, resulting in a greater momentum flux convergence over the equatorial Indian Ocean with warming. These changes can be explained using simple vorticity arguments. Since the
large-scale summertime circulation is in Sverdrup balance, larger troposphere-stratosphere mass flux due to tropopause uplift leads to greater divergence and, consequently, stronger rotational flow. This elegant mechanism is also qualitatively reproduced in a simple aquaplanet general circulation model. Given that the Asian summer anticyclone plays a vital role in the breaking of Rossby waves in the vicinity of the subtropical tropopause (Postel and Hitchmann, 1999), these structural changes may influence the frequency and intensity of such events as well as stratosphere-troposphere tracer and mass exchange (Chen,
1995; Dunkerton, 1995). Further, as pointed out by Postel and Hitchmann (1999), the Asian summer anticyclone appears as a smooth large-scale system in monthly and seasonal averages. In reality, it is a highly dynamic system that shows variability on finer spatial and temporal scales (Siu and Bowman, 2020), closely associated with the background flow (Rupp and Haynes, 2021). Given this background, the response of the Asian monsoon anticyclone to warming is a worthy candidate that merits detailed investigation and will be taken up as future work.

During winter, the documented weakening of stationary waves (Joseph et al., 2004; Wills et al., 2019) causes a weakening of the upper tropospheric eddy westerly winds over the East Pacific. This almost shuts off the equatorward propagation of wave activity, leading to a decrease in the magnitude of the zonal deceleration experienced by this region. We also note a drastic drop in equatorward stationary and transient potential vorticity fluxes on both sides of the equator. Together, these changes cumulatively indicate a decrease in the intrusions of high potential vorticity air and tracers into the deep tropics (Waugh and
Polvani, 2000). Such changes in the wave-mean flow interactions over the East Pacific bear important consequences for the weather and climate variability of this region. Indeed, the changes observed in eddy momentum fluxes, wave activity, and potential vorticity fluxes, as suggested by the CMIP6 projections, are a manifestation of many complicated processes induced by warming (Shaw, 2019; Vallis et al., 2015). In fact, warming-induced changes along similar lines have been reported previously as well. Freitas and Rao (2014) show that stationary Rossby wave propagation over long distances becomes unfavorable for the



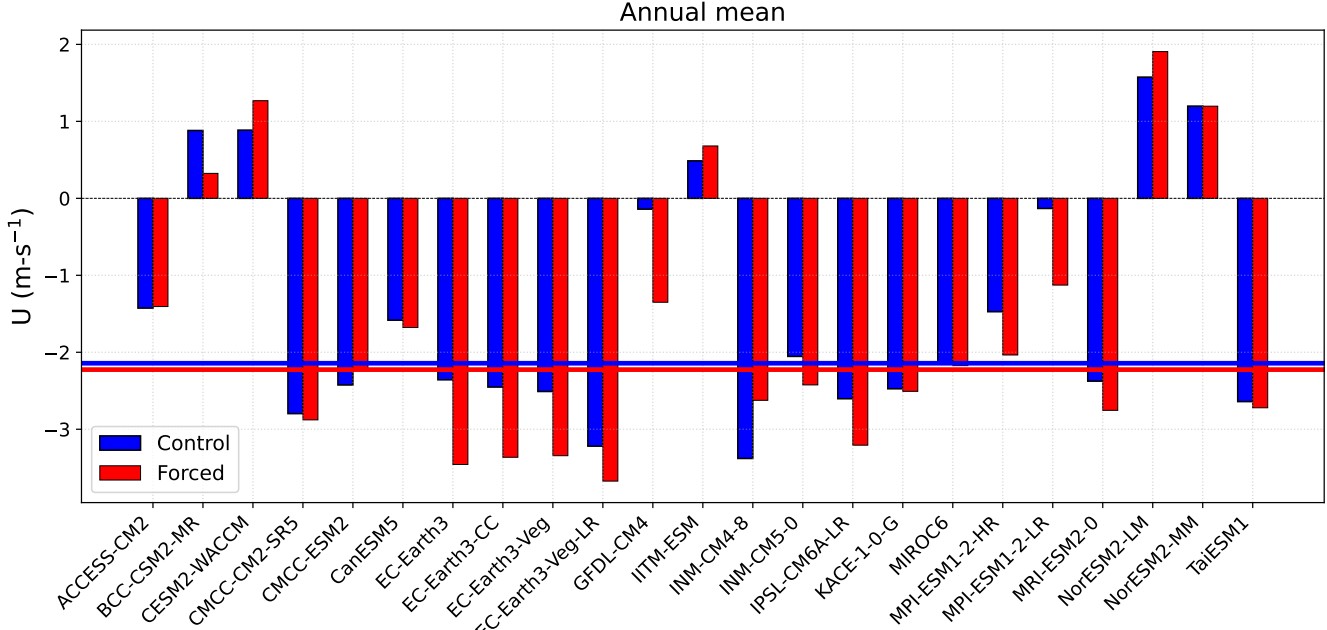

**Figure 14.** Spread of the annual mean upper tropospheric zonal mean zonal wind in the control (blue) and forced (red) ensembles, averaged over $\pm 5°$ and 150-300 mbar. The horizontal lines indicate the median values of the forced (red) and control (blue) ensembles.

main waveguides in the NH and SH. Simpson et al. (2016) show that changes in the character of intermediate-scale stationary waves cause the circulation changes that impact the precipitation over California and the eastern Mediterranean. However, the warming-induced changes noted over the Central and Eastern Pacific are at odds with the recent trends in reanalysis data, which suggest an increase in the PV intrusions over this area (Nath et al., 2017). Such trends in observations have been attributed to the strengthening of the equatorial westerly window and, more fundamentally, the strengthening trend of the equatorial SST

gradient. Therefore, this disagreement between observational and simulated trends originates in the mismatch in equatorial Pacific SST gradient trends, usually attributed to model biases or internal climate variability (Seager et al., 2019; Watanabe et al., 2021; Lee et al., 2022).

Similar discrepancies between observed and modeled trends have been shown to exist in other quantities related to the tropical general circulation. Santer et al. (2005) show that multi-decadal trends of tropospheric temperature amplification, captured

faithfully by climate models, was observed in only one observational dataset. Mitas and Clement (2006) show that models and reanalyses disagree on the thermodynamic trends as well; while models exhibit warming in the upper troposphere and increased static stability, reanalyses show a cooling trend with decreased static stability. Further, models and reanalyses show opposite trends in the strength of the NH Hadley circulation (Chemke and Polvani, 2019). This stems from poor representation of diabatic heating (Chemke and Polvani, 2019) and long-term climate variability (Zaplotnik et al., 2022). Such disagreements

are a major source of uncertainty in future projections, and present a significant challenge to the scientific community.





Bringing together the changes in the upper troposphere with warming, all terms involved in the momentum budget decrease in magnitude. The net effect on the annual and zonal mean zonal flow is a very slight decrease in the upper-tropospheric easterlies (compare median values of the control and forced ensembles in Figure 14). This trend also holds for the seasonal mean values (Figure S5). Indeed, there is a fair spread amongst the models; some models in the ensemble indicate a switch with equatorial superrotation as an outcome of climate change, while others remain superrotating in both the control and forced sets (Figures 14 and S5). However, the marginal decrease in the median values of the zonal mean zonal wind suggests that although projections suggest that the magnitudes of both the eddy and mean terms decrease with warming, the upper tropospheric balance of momentum may not change significantly.

In all, one aspect of this study showcases the extent of internal compensations that occur when upper equatorial momentum fluxes are studied in a zonal mean sense. By presenting the results for the control simulations of CMIP6 alongside those for reanalysis, we hope to aid modeling groups in closing the gap between the two and, perhaps, provide better future projections on global and regional scales (Hall, 2014; Xie et al., 2015). For example, the smaller magnitude of eddy acceleration in the summer in the control set, especially in the CP-WA region, as compared to present-day estimates, possibly points to issues with the organization of the tropical convergence zone in the model runs. In summer, differences in the eddy and mean meridional momentum flux in the CP-WA region result in weaker upper-tropospheric easterlies in the CMIP6 control than in reanalysis. The second set of results involves state-of-the-art projections suggesting significant changes to the dominant terms contributing to the upper tropical momentum budget. Indeed, both the mean meridional and eddy fluxes are seen to change in a statistically significant manner. Particular attention was drawn to stationary contributions that dominate the eddy fluxes. At present, the balance of these terms is oriented to give upper tropical zonal mean easterlies (Lee, 1999; Dima et al., 2005). Which way this balance tilts will affect the future state of the atmosphere, including potential tipping to superrotation as is thought to have prevailed in past climates (Pierrehumbert, 2000; Tziperman and Farrell, 2009).





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
