# Peer review of "Changes in the tropical upper tropospheric zonal momentum balance due to global warming"

_EGUsphere, 2023_

## Author Response (AR1)

**Response to Reviewer1**

(Please note that the Reviewer's comments are in italicized font and our responses are in normal font)

**General comments**

*The authors evaluate the upper tropospheric momentum budget for ERA-5 reanalysis, and CMIP6 historical and SSP585 simulations. The paper explores the magnitudes of the zonal mean state advection of absolute vorticity and the stationary waves throughout the year, and across longitudes. In wintertime, over the East Pacific, they find that under the SSP585 scenario, the stationary waves contribution strongly decreases, likely due to the future El Niño SST anomalies simulated in CMIP6 but not currently observed.*

*The paper is clearly written and structured, presents some useful analysis of reanalysis and model data, the methods generally seem appropriate, and the results support the conclusions.*

**Answer:** Thank you for your critical reading of our manuscript. We have updated the manuscript as per your suggestions and provided a point-by-point response to your comments below.

**Specific comments**

***Budget closure****: The ERA5 momentum budget shows a large summertime residual of similar magnitude to the eddy term. This is discussed in the text, but would it be possible to either compare closure against other studies using ERA5, or to look into the cause of this? For example, ERA5 does provide the 'mean eastward wind tendency due to parametrisations' on model levels, alongside code to interpolate this to pressure levels. Given the focus of the paper, this seems an important point to have some explanation for. Could it arise from sub-daily transient activity? How well does the budget close across the pressure level for JJA?*

**Answer:**

The large residual was of concern to us too, and in fact, its large magnitude during the summer is somewhat well known, as we discuss in the manuscript. Moreover, the residual is quite large during NH summer than NH winter, as also pointed out in previous work (please compare Figures 1f & 2f in Lin et al. [2008] and Figure 4a & 4b in Yang et al. [2013]). As we discuss in the manuscript, the usual suspect here is Convective Momentum Transport [CMT; Carr and Bretherton, 2001, Lin et al., 2008, Yang et al., 2013].

Further, the following excerpt from Yang et al. [2013] sheds some more light on the issue at hand, "It should also be mentioned here that the estimated CMT is quantitatively only able to account less than half of the whole residual term even over the oceans, reflecting either the crudeness of the representation of the cloud detrainment rate by the precipitation rate or the contamination of X by other sources (e.g., gravity wave activities and errors of data itself)."

Indeed, our usage of daily averaged data underestimates the fluxes slightly. On repeating the exercise using 6-hourly data at a finer horizontal resolution of $1°$, the peak fluxes during the monsoon are slightly higher than our original result (compare Figure 1 below with Figure 1 in the main document). However, the residual obtained in this exercise is similar to what we obtain using daily-averaged data at $2.5°$.

Repeating the exercise using daily averaged ERA-Interim data at $2.5°$ gave a much smaller residual (Figure 1). This suggests that the relatively larger residual is possibly an ERA5 artifact. Per your

suggestions, the vertical profile of the budget terms (Figure 2) also suggests that the residual is small compared to other terms. We have added this Figure to the Supplementary material as Figure S1.

In this context, we have added the following to the text.

lines 174-177

"Indeed, the use of daily averaged wind fields underestimates the eddy covariances; however, the same calculation using the ERA-Interim dataset resulted in a relatively smaller residual (not shown), indicating that the large residual here may be an ERA5 artifact. However, the budget terms in the pressure-latitude plane averaged over the boreal summer season suggest that the residual is much smaller than the horizontal eddy and mean flow terms (Figure S1)."

Given that the results and discussion in the manuscript surround the eddy and mean fluxes, we feel that further discussion of the residual may be beyond the scope of the present work but merits a separate investigation.

*Novelty: The paper is well-contextualized in terms of the previous literature, and I believe presents some new and interesting results. The authors highlight that changes in the regional eddy fluxes have not received much attention. However, it would be helpful to promote more clearly in the abstract, introduction and conclusions the key new findings in this study, perhaps simply by rephrasing from the passive to active voice: 'we find'.*

**Answer:** Thank you. To promote our new findings, we have made changes to the voice in the Abstract, Introduction, and Conclusions.

*CMIP6 simulations: r1i1p1f1 is not the main simulation for some centres, and its meaning is not standard across centres. It may be possible to expand the number of modules used, although the 23 selected seem to cover a reasonable range of modelling centres.*

**Answer:** We chose *r1i1p1f1* because there was no reason for us to explicitly prefer any other variant-id. In fact, in an earlier analysis (https://arxiv.org/abs/2107.09646), we used *piControl* and *1pctCO2* and obtained similar results.

**Methods**

*Which pressure levels are used in the CMIP6 data, do these match the ERA5 levels used?*

**Answer:** We have now included these details in the Data section. Please see lines 95 and 109-111.

*wap is not listed in the CMIP6 table, were vertical fluxes not evaluated for CMIP6?*

**Answer:** No, vertical fluxes were not calculated for CMIP6 dataset. Using ERA5 reanalysis, we identified that the horizontal mean and eddy fluxes are the largest contributors to the zonal mean balance of momentum. In the context of climate change, we evaluate the changes to these horizontal fluxes. This approach streamlined and simplified our analysis significantly.

*The acronyms (ua, va, tos, etc.) for the CMIP6 variables given in the table are not explained*

**Answer:** We have added the variable descriptions in the Data section. Please see line 107-109.

*In section 2.1.3 it would be useful to note that the simulation includes a stationary wave SST, and to*

[Figure]

Figure 1: (top) Same as Figure 1 of the manuscript except for six hourly ERA5 data at 1° horizontal resolution. (bottom) Same as Figure 1 of the manuscript except for daily averaged ERA-Interim data at 2.5° horizontal resolution.

*refer to the supplementary information, rather than relying on knowledge of the paper referenced.*

**Answer:** We have added the description. We have also brought the relevant figure from the Supplementary into the main document (see Figure 11).

lines 119-121

"To simulate the large-scale monsoon flow, the control run is forced using a stationary wave-1 SST perturbation centered at 30°N superimposed on a zonally symmetric SST distribution shifted to 10°N [Wu and Shaw, 2016], along with perpetual July insolation."

*Calculating the fluxes from daily means appears to give reasonable results. It could be noted that*

[Figure]

Figure 2: Climatological latitude-pressure profile of all terms in the zonal mean zonal momentum budget averaged over JJA.

*this excludes any short-lived, subdaily activities.*

**Answer:** We have added the note in the text.

lines 133-134

"Since we use daily-averaged data to compute the fluxes, our calculation excludes short-lived, high-frequency activities."

*Fig 2/averaging regions: I initially found the averaging regions described in the text hard to interpret, vertical lines on Fig 2 could make these clearer.*

**Answer:** We have updated the Figure accordingly.

*Line 283: "A possible reason for these discrepancies is that the model fluxes tend to be slightly displaced than those for present-day reanalysis." Not clear to me what is meant here, is this referring to spatial displacement, and in what sense?*

**Answer:** Yes, we agree that the sentence creates more confusion than it brings clarity. We have removed that sentence.

*Line 308: First sentence here refers to Fig 6, should note that this is discussing summer.*

**Answer:** Corrected.

*Fig 10: There is a shift to look at the Northern Hemisphere only here, is there a reason for this?*

**Answer:** Yes. This is following the discussion in Section 4.2. As can be seen by comparing Figures 9b and d, the lower stratospheric eddy fluxes are strengthened over the Asia-Africa region. We hypothesized that this is due to the increased subtropical mass flux into the stratosphere during the NH summer, plausibly from the Indian summer monsoon flow. Further, the Asian summer anticyclone plays a vital role in the breaking of Rossby waves in the vicinity of the subtropical tropopause [Postel and Hitchmann, 1999]; structural changes induced by warming may influence the frequency and

intensity of such events as well as stratosphere-troposphere tracer and mass exchange [Chen, 1995, Dunkerton, 1995].

*Line 386: "Quite clearly, comparing the balance for the two scenarios suggests a higher degree of compensation between the beta and stretching terms via a larger divergence in the forced ensemble than the control set." This is not clear to me from Fig 10, the residual between the two terms may need to be shown to support this.*

**Answer:** We have added the full vorticity budget [Eq 4 in the manuscript Sardeshmukh and Held, 1984] in the Supplementary.

*Fig. 11: Not fully clear what is shown here. Do a and b show a vertical slice of the horizontal streamfunction, or a lat-pressure streamfunction?*

**Answer:** Yes, the Figure shows a vertical slice of the horizontal streamfunction.

*Fig. 12 caption: it needs to be made clear in the caption that quivers show differences in left column but absolute values in right column.*

**Answer:** We have added that note in the Figure caption.

"Please note, in panel d the quivers are for the forced run rather than the difference between forced and control simulations as in panel c."

*Fig. 13 caption: please could you specify that positive values correspond to northward fluxes in both panels, to make this simpler to interpret.*

**Answer:** We have added that note in the Figure caption.

"In both panels, positive (negative) values correspond to northward (southward) fluxes."

*Line 446: It would be helpful to note the longitudes to look at for the westerly duct here, particularly given the change in longitude axis from -180-180 to 0-360.*

**Answer:** We have marked out the longitudes to look for with a box.

*Line 534-535: I found this sentence confusing. To me, "captured faithfully" implies models are consistent with the observations, but the sentence continues to say they are not. Is the intention to say that models consistently show this behavior?*

**Answer:** We have removed the word "faithfully"

*Line 544-545: "Indeed, there is a fair spread amongst the models; some models in the ensemble indicate a switch with equatorial superrotation as an outcome of climate change" From Fig 14 I can't see any that go from -ve in the control to +ve in the forced, as seems implied by this sentence.*

**Answer:** We have changed the sentence.

lines 554-555

"some models in the ensemble remain superrotating while others remain subrotating in both the control and forced sets"

**Technical corrections**

*Equations 1 & 3 use x and y and lambda and phi respectively. Similarly deviations from the zonal*

*mean are denoted by asterisks and primes differently in these equations. It would be good to make these consistent.*

**Answer:** Corrected.

*Section 2.2.3: should F by $F_s$ in the sentence discussing WKB theory?*

**Answer:** Yes. Corrected.

*Line 225: "While the wave activity flux in (arrows in Figure 4b) captures stationary contributions" Delete 'in'*

**Answer:** Corrected.

*Line 339: missing &*

**Answer:** Corrected.

*Fig. 5: Please add a legend to avoid readers scrolling to check Fig. 3.*

**Answer:** Figure updated, as suggested.

*Fig. 9: Could the same colorbar be used for both columns?*

**Answer:** Yes.

**References**

Matthew T Carr and Christopher S Bretherton. Convective momentum transport over the tropical pacific: Budget estimates. *Journal of the atmospheric sciences*, 58(13):1673–1693, 2001.

P Chen. Isentropic cross-tropopause mass exchange in the extratropics. *Journal of Geophysical Research*, 100:16661–16673, 1995. doi: 10.1029/95JD01264.

TJ Dunkerton. Evidence of meridional motion in the summer lower stratosphere adjacent to monsoon regions. *Journal of Geophysical Research*, 100:16675–16688, 1995. doi: 10.1029/95JD01263.

Jia-Lin Lin, Brian E Mapes, and Weiqing Han. What are the sources of mechanical damping in matsuno–gill-type models? *Journal of Climate*, 21(2):165–179, 2008. doi: 10.1175/2007JCLI1546.1.

GA Postel and MH Hitchmann. A climatology of rossby wave breaking along the subtropical tropopause. *Journal of the Atmospheric Sciences*, 56:359–373, 1999. doi: 10.1175/1520-0469(1999)056<0359:ACORWB>2.0.CO;2.

Prashant D Sardeshmukh and Isaac M Held. The vorticity balance in the tropical upper troposphere of a general circulation model. *Journal of Atmospheric Sciences*, 41(5):768–778, 1984. doi: 10.1175/1520-0469(1984)041<3C0768:TVBITT>2.0.CO;2.

Yutian Wu and Tiffany A Shaw. The impact of the asian summer monsoon circulation on the tropopause. *Journal of Climate*, 29(24):8689–8701, 2016. doi: 10.1175/JCLI-D-16-0204.1.

Wenchang Yang, Richard Seager, and Mark A Cane. Zonal momentum balance in the tropical atmospheric circulation during the global monsoon mature months. *Journal of the atmospheric sciences*, 70(2):583–599, 2013.

**Response to Reviewer 2**

(Please note that the Reviewer's comments are in italicized font and our responses are in normal font)

**Summary**

*The authors diagnose the zonally resolved zonal momentum budget for the tropics in reanalysis data and CMIP models, both control and SSP8.5-forced. They split out the contributions from many different terms, including the rotational and divergent components.*

*The writing is for the most part clear, and the Introduction sets up a potentially compelling story. But—and this might just be me—I struggled to keep focus/interest once the manuscript got into its results. The first few figures, there were already a number of things regarding their interpretation that were tripping me up, as noted below. So it was then hard to then make much of the remainder of the results.*

*That said, I don't see any glaring errors in the analyses, and the results presented are worthwhile contributions to the literature pending some cleaning up as detailed below.*

**Answer:** Thank you for the comments on our manuscript. We hope that the revision makes the presentation a little clearer and keeps the reader engaged through the manuscript.

**Major Comments**

**** Residual term and daily data** *In your Fig. 1, in NH summer the leading balance is not, as you state, a two-term balance between the eddy momentum flux divergence and mean meridional advection...it's a three-term balance of those and the residual. In your discussion of this issue, you don't bring up what to me seems like a likely contributor: your use of daily averaged data. The standard is to use 6-hourly (or 3-hourly) covariances of instantaneous fields.*

*I'm not necessarily saying you need to re-do the whole thing with hourly data, as that's a heavy lift indeed. But it would make everything much more compelling, and ERA5 hourly data is available to make this possible.*

**Answer:**

Indeed, our usage of daily averaged data underestimates the fluxes slightly. On repeating the exercise using 6-hourly data at a finer horizontal resolution of 1°, the peak fluxes during the monsoon are slightly higher than our original result (compare Figure 1 below with Figure 1 in the main document). However, the residual obtained in this exercise is similar to what we obtain using daily-averaged data at 2.5°.

Further, repeating the exercise using ERA-Interim using daily averaged data at 2.5° gave a much smaller residual (Figure 1). This suggests that the larger residual is possibly an ERA5 artifact. Per your suggestion, we have repeated the calculation using 3-hourly 0.25° resolution ERA5 data (Figure 2); however, the results are largely similar to the one we have presented in the manuscript and not much different from that calculated using coarser 1° 6-hourly data.

We would like to add that the residual is quite large during NH summer than NH winter, as pointed out in previous work (please compare Figures 1f & 2f in Lin et al. [2008] and Figure 4a & 4b in Yang

et al. [2013]). As we discuss in the manuscript, the usual suspect here is Convective Momentum Transport [CMT; Carr and Bretherton, 2001, Lin et al., 2008, Yang et al., 2013].

The following excerpt from Yang et al. [2013] sheds some more light on the issue at hand, "It should also be mentioned here that the estimated CMT is quantitatively only able to account less than half of the whole residual term even over the oceans, reflecting either the crudeness of the representation of the cloud detrainment rate by the precipitation rate or the contamination of X by other sources (e.g., gravity wave activities and errors of data itself)."

Further, following the suggestions of Reviewer 1, we have also looked at the latitude-height sections of the zonal mean budget terms averaged over JJA (Figure S1). The residual is quite small here in comparison to the horizontal eddy and mean terms.

In this context, we have added the following to the text.

lines 173-178

"Indeed, the use of daily averaged wind fields underestimates the eddy covariances; however, the same calculation using the ERA-Interim dataset resulted in a relatively smaller residual (not shown), indicating that the large residual here may be an ERA5 artifact. However, the budget terms in the pressure-latitude plane averaged over the boreal summer season suggest that the residual is much smaller than the horizontal eddy and mean flow terms (Figure S1)."

**Stationary vs. transient eddies**

*The framing is in terms of the mean meridional circulation on the one hand versus all eddies on the other hand. But stationary eddies and transient eddies are very different from one another, and you argue that one or the other plays more important roles in different locations and contexts. So I'm wondering if it's worth the effort to explicitly disentangle them, presenting results for both individual eddy terms.*

**Answer:**

We had considered this partition. Our objective was to highlight the contrasting nature of the eddies in the tropical momentum balance – that both tropical and extratropical modes are involved in the balance rather than just tropical modes. We found that this was most succinctly captured by partitioning the tropics and then using the rotational-divergent partition. We felt it would be difficult to establish the involvement of tropical and extratropical modes using the stationary/transient partition because both tropical and extratropical modes are composed of stationary and transient components.

As you can see in Figure 3, the eddy momentum flux convergence by stationary eddies explains a large portion of the total eddy momentum flux – a fact that holds for the Asia-Africa region as well because the climatological stationary Rossby waves are prevalent there. However, the transient contribution is relatively small. In comparison, the eddy momentum flux convergence associated with the Af-A and CP-WA regions is much larger (compare with Figure 3a in the manuscript).

Further, there is a fairly large literature surrounding stationary and transient eddies [for example Dima et al., 2005, Zurita-Gotor, 2019]. However, to the best of our knowledge, the prominence of stationary Rossby waves in the Asia-Africa region versus the seasonally sensitive prominence of the extratropical waves in the East Pacific hasn't received much attention.

**Minor comments**

*- On superrotation, see also Zhang and Lutsko, doi.org/10.1175/JAS-D-22-0066.1*
**Answer:** We have included the reference.

[Figure]

Figure 1: (top) Same as Figure 1 of the manuscript except for six hourly ERA5 data at 1° horizontal resolution. (bottom) Same as Figure 1 of the manuscript except for daily averaged ERA-Interim data at 2.5° horizontal resolution.

lines 70-71

"Recently, it has been shown that stationary eddies in the tropical upper troposphere drive a seasonal superrotation, not seen in the annual mean [Zhang and Lutsko, 2022]."

*- On the SSP8.5 scenario, just be aware that it is now virtually certain to not occur: doi.org/10.1038/d41586-020-00177-3*

**Answer:** Right, we have made a note in the text.

lines 104-106

"It must be mentioned that the SSP5-8.5 scenario is now practically implausible [Hausfather and

[Figure]

Figure 2: Same as Figure 1 of the manuscript except for three hourly ERA5 data at 0.25° horizontal resolution.

[Figure]

Figure 3: Partitioning of the full eddy momentum flux convergence (blue) into contributions by stationary (red) and transient (orange)) eddies. All quantities are averaged over the 150-300 mbar layer.

Peters, 2020]; however, we use it here as an extreme case of anthropogenic warming."

*- Table 1 :: there's no need for the variable columns, since every single one is checked. Just list the model names and note either in the caption or main text that they all include the five variables you've listed.*

**Answer:** Corrected.

[Figure]

Figure 4: All components of the rotational divergent partition of the zonal mean eddy momentum flux convergence.

*- Text after Eq. 3, the F term is missing the "s" subscript*

**Answer:** Corrected.

*- L147 :: if it's already been published and discussed, it's not "surprising"*

**Answer:** Corrected.

*- Fig. 3:: If you're saying the other components that aren't plotted are small, then why doesn't the total visually add up to the sum of the plotted ur*vd and ur*vr lines in a lot of places?*

**Answer:** Please refer to Figure 4 in this document. The terms $u_d v_d$ and $u_d v_r$ are much smaller than the other terms and are fairly constant throughout the year. We have added the following sentence in the main text

lines 220-221

"Please note that the contribution from the $u_d^* v_d^*$ and $u_d^* v_r^*$ terms are much smaller than the other terms and are fairly constant throughout the year (not shown)."

*- Fig. 3b :: the ur*vd legend label doesn't show up as dash-dotted, just as a shorter line than the total*

**Answer:** We have updated the Figure. The dash-dotted $u_r v_d$ line plots are now dashed.

*- L259 :: "As long as ur-vd or tropical features dominate the eddy and mean fluxes, they should oppose each other in strength and symmetry" I don't understand this*

**Answer:** We have removed the sentence.

*- "tropical momentum balance is delicate" this is said a few times, but what does it really mean? Regarding the response to a forcing, the prevailing balance does not constrain the response in any*

*way—a given forcing could operate primarily by a third term for example, so I don't follow e.g. L488-491.*

**Answer:** Right. Warming-induced changes to the zonal momentum balance may affect the terms involved in different ways. We have restructured these bits for more clarity.

lines 499-501

"Given that the tropical momentum balance is delicate on both geographical and seasonal scales, changes in the diabatic heating distribution associated with changing precipitation patterns may change the balance of momentum and, therefore, the zonal mean zonal winds as well."

**References**

Matthew T Carr and Christopher S Bretherton. Convective momentum transport over the tropical pacific: Budget estimates. *Journal of the atmospheric sciences*, 58(13):1673–1693, 2001.

Ioana M. Dima, John M. Wallace, and Ian Kraucunas. Tropical zonal momentum balance in the NCEP reanalyses. *Journal of the Atmospheric Sciences*, 62(7):2499–2513, jul 2005. doi: 10.1175/ jas3486.1.

Zeke Hausfather and Glen P Peters. Emissions–the 'business as usual'story is misleading. *Nature*, 577(7792):618–620, 2020.

Jia-Lin Lin, Brian E Mapes, and Weiqing Han. What are the sources of mechanical damping in matsuno–gill-type models? *Journal of Climate*, 21(2):165–179, 2008. doi: 10.1175/ 2007JCLI1546.1.

Wenchang Yang, Richard Seager, and Mark A Cane. Zonal momentum balance in the tropical atmospheric circulation during the global monsoon mature months. *Journal of the atmospheric sciences*, 70(2):583–599, 2013.

Pengcheng Zhang and Nicholas J Lutsko. Seasonal superrotation in earth's troposphere. *Journal of the Atmospheric Sciences*, 79(12):3297–3314, 2022.

Pablo Zurita-Gotor. The role of the divergent circulation for large-scale eddy momentum transport in the tropics. part i: Observations. *Journal of the Atmospheric Sciences*, 76(4):1125–1144, apr 2019. doi: 10.1175/jas-d-18-0297.1.

---

## Author Response (AR2)

**Response to Editor**

(Please note that the Reviewer's comments are in italicized font and our responses are in normal font)

*The authors have done a generally good job at responding to the reviewers comments in the first round. However, both reviewers raised as a major comment the lack of budget closure in ERA5, and I am not sure that this is adequately addressed.*

*As editor, I therefore suggest the authors further address this point in the following ways:*

We thank the Editor for giving us the opportunity to improve our manuscript.

*- One reviewer points out that the ERA5 dataset includes the analysis increments and sub-gridscale accelerations. Is there a reason that this data cannot be obtained to test whether this is the cause of the discrepancy?*

**Answer:** We thank the Reviewer for this comment and the Editor for pointing out that the comment was unanswered in the Response. The 'Mean eastward wind tendency due to parameterisations' is available as a forecast category variable in the ERA5 catalog rather than the assimilated quantities used in the manuscript. Since a direct comparison cannot be made between assimilated and forecasted quantities, a meaningful interpretation of our results cannot be made using the suggested variable. Another issue is that the parameterisation tendency variable is on model levels, as pointed out by the reviewer as well. However, the code provided by ECMWF to interpolate this to pressure levels requires one to download data on all 137 model levels. The size is 125MB for a single day's data; this becomes $\sim 1.8$ TB for 40 years for a single variable, and converting this to pressure levels would be quite time-consuming and resource-intensive. Again, since a direct interpretation of assimilated quantities cannot be facilitated using forecast class quantities, we did not do this exercise.

*- The authors suggest that convective momentum transport is unlikely to be the reason for the imbalance, but it is hard to see what else it could be. Perhaps examining the latent heating rate in ERA5 would indicate whether the region of strong imbalances is a region of strong deep convection?*

**Answer:** The diabatic heating rate suggests that it is indeed convective momentum transport in the regions of deep convection in the tropics that contributes to the momentum budget residual, as can be seen in Figure 1. We have now modified the third paragraph of Section 3.1 to incorporate this and added the Figure 1 to the Supplementary Material.

Lines 168-181

"The residual ($X$ in Figure 1) attains a large non-negligible value during the summer ($\sim 0.5 \, \mathrm{ms}^{-1}\mathrm{day}^{-1}$) and is much stronger than the other remaining terms. This may be due to intense convective momentum transport that is known to occur over the Indian Ocean - Maritime Continent region during this season [Lin et al., 2008, Yang et al., 2013]. However, convective momentum transport is usually modeled as a vertical convergence of zonal eddy momentum flux [Carr and Bretherton, 2001, Lin et al., 2008, Yang et al., 2013], which is an explicit forcing term in our momentum equation (Equation 1; green dashed curve in Figure 1). A comparison of the spatial distributions of $X$ and the diabatic heating rate (Figure S1) suggests that convective momentum transport by unresolved eddies is an important contributor to the budget residual $X$ in the deep convective regions over the open oceans in the tropics. Indeed, the use of daily averaged wind fields underestimates the eddy covariances; however, the same calculation using the ERA-Interim dataset [Dee et al., 2011] resulted in a relatively smaller residual (not shown), indicating that the large residual here may be an ERA5 artifact.

[Figure]

Figure 1: JJAS and 150-300 mbar averaged (top) Momentum budget residual (unit ms$^{-1}$day$^{-1}$) and (bottom) diabatic heating rate (unit K day$^{-1}$) estimated as a residual from the thermodynamic energy equation [Holton and Hakim, 2012].

However, the budget terms in the pressure-latitude plane averaged over the boreal summer season suggest that the residual is much smaller than the horizontal eddy and mean flow terms (Figure S2). The residual term weakens after the monsoon and becomes comparable in magnitude to the other terms that remain small throughout the year. Given the systematic dominance of the mean meridional advection and eddy momentum flux convergence throughout the year, the discussion that follows will be focused entirely on these two terms."

*- The CMIP6 multi-model mean also seems to show a fairly large residual. Is this of a similar magnitude and spatial pattern as the one for ERA5? Could this be further evidence that the missing processes are subgrid-scale model terms?*

**Answer:** Thank you for the comment. We would like to clarify that we have not explicitly calculated the full momentum budget for the CMIP6 models. As we explained in our response to Reviewer 1 as well as in lines 287-289 of the manuscript, having identified the dominant terms using the ERA5 reanalysis, we have only evaluated the warming-induced changes to these terms. This approach streamlined and simplified our analysis significantly. Indeed, the CMIP models suffer from their own set of biases, and their manifestation in the budget residual could be an avenue for future investigation.

*I think the paper is overall sound, and a bit more discussion of this point will allow the reader to assess the budget and any impacts of the imbalance appropriately.*

*Also note that the WCD policy on data availability (`https://www.weather-climate-dynamics. net/policies/data_policy.html`) requires authors to make their data available in a publicly accessible repository. Please ensure that either the simulation outputs, or information (namelists, config files) sufficient to reproduce the simulations are placed in a public repository and made available.*

**Answer:** Thank you. We have included the details of the publicly accessible repositories.

**Minor comments**

*Line 34: remove "however"*

*Line 130: Specify that the last term is the "X" term.*

*Libe 145: Rossby waves -> Rossby wave*

*Line 386: This is a bit confusing because you earlier talk about a decrease in the tropical mass flux. I assume here you are talking about mass flux through the tropopause, and that is the difference, but it might make sense to note this explicitly*

**Answer:** Thank you. Corrected.

**References**

Matthew T Carr and Christopher S Bretherton. Convective momentum transport over the tropical pacific: Budget estimates. *Journal of the atmospheric sciences*, 58(13):1673–1693, 2001.

D. P. Dee et al. The era-interim reanalysis: configuration and performance of the data assimilation system. *Quarterly Journal of the Royal Meteorological Society*, 137(656):553–597, 2011. doi: 10.1002/qj.828.

JR Holton and G Hakim. *An Introduction to Dynamic Meteorology*. Academic Press, New York, 5 edition, 2012.

Jia-Lin Lin, Brian E Mapes, and Weiqing Han. What are the sources of mechanical damping in matsuno–gill-type models? *Journal of Climate*, 21(2):165–179, 2008. doi: 10.1175/2007JCLI1546.1.

Wenchang Yang, Richard Seager, and Mark A Cane. Zonal momentum balance in the tropical atmospheric circulation during the global monsoon mature months. *Journal of the atmospheric sciences*, 70(2):583–599, 2013.